# Making Look-Ahead Active Learning Strategies Feasible with Neural Tangent Kernels

**Mohamad Amin Mohamadi**[*]
University of British Columbia
lemohama@cs.ubc.ca

**Wonho Bae**[*]
University of British Columbia
whbae@cs.ubc.ca

**Danica J. Sutherland**
University of British Columbia
Alberta Machine Intelligence Institute
dsuth@cs.ubc.ca

## Abstract

We propose a new method for approximating active learning acquisition strategies that are based on retraining with hypothetically-labeled candidate data points. Although this is usually infeasible with deep networks, we use the neural tangent kernel to approximate the result of retraining, and prove that this approximation works asymptotically even in an active learning setup – approximating "look-ahead" selection criteria with far less computation required. This also enables us to conduct sequential active learning, i.e. updating the model in a streaming regime, without needing to retrain the model with SGD after adding each new data point. Moreover, our querying strategy, which better understands how the model's predictions will change by adding new data points in comparison to the standard ("myopic") criteria, beats other look-ahead strategies by large margins, and achieves equal or better performance compared to state-of-the-art methods on several benchmark datasets in pool-based active learning.

## 1 Introduction

Deep learning has drastically advanced over the past decade along with a dramatic increase in the volume of available data. It is, however, time-consuming and expensive to manually annotate large datasets. Active learning attempts to alleviate this by allowing a model to "actively" request annotation of specific data points, with the expectation that a model trained on informative points will learn a better prediction than one on a random, "passively" labeled set of the same size.

The most common form of active learning is based on using an *acquisition function* to measure the informativeness of potential data points or batches from some unlabeled pool. Many successful acquisition functions are based on the current model's uncertainty, such as maximum entropy [1] and Bayesian Active Learning by Disagreement (BALD) [2]. These functions are based on the expectation that training on points about which the current model is uncertain will be effective at decreasing the future model's uncertainty about similar points. This assumption, however, often does not hold: uncertain points might be inherently difficult to predict (aleatoric uncertainty) or simply too difficult for the model to learn right now.

It would be helpful, then, to see how a new data point would change a model, potentially with respect to other unseen data. One such strategy is known as Expected Model Change, which approximates

---

[*]These authors contributed equally.

36th Conference on Neural Information Processing Systems (NeurIPS 2022).

how much a model's parameters will change when observing a new hypothetically-labeled data point [3, 4]. This approach, though, only considers the magnitude of change in parameters, not the model's actual outputs on the data distribution, and it only looks at the size of the first stochastic gradient (SGD) step with the new data point rather than considering the full trajectory of training.

We will refer to acquisition strategies which consider full retraining of a model based on hypothetical observations as *look-ahead* criteria. Previous attempts to use look-ahead criteria based on the change in model output include Expected Error Reduction [5] and Expected Model Output Change [6]. These approaches have been used only with specialized models such as Naïve Bayes and Gaussian processes, where retraining on every new candidate data point is feasible. These classes of models, however, tend to not work as well as modern neural networks, meaning these approaches are outperformed by simpler acquisition functions on stronger models.

In this work, we propose an algorithm that makes it feasible to conduct active learning with look-ahead acquisition strategies using general neural networks. More specifically, we approximate a network using its Neural Tangent Kernel (NTK) [7], which makes it possible to obtain the behavior of retraining a network on a candidate data point without actually retraining the network. At each query step, we use the local approximation of the neural network as a proxy to compute a look-ahead acquisition function, from which we query a new data point. We prove that in the asymptotic regime where the width of the neural network goes to infinity, this approximation agrees with the result of actually retraining the network, even in an iterative setup such as in active learning. Our approximation decreases the wall-clock time more than $100$ times compared to naïve computation of a look-ahead acquisition function, making them far more feasible in practice.

Our proposed method outperforms existing look-ahead strategies by large margins, and achieves equal or better performance compared to the state-of-the-art methods on several benchmark datasets in pool-based active learning, including MNIST [8], SVHN [9], CIFAR10 [10], and CIFAR100 [10]. The NTK approximation also makes it possible to decouple receiving new data labels and SGD training, unlike previous active learning methods, where knowing true labels does not add any information without SGD training. This is useful when annotation is fast (yet expensive) but model training is slow, since adding data to an existing NTK approximation is far faster than retraining with SGD. Sequentially adding true labels of new data gives performance substantially better than more common batch setups.

## 2 Preliminaries

**Active learning.** In pool-based active learning, we have a labeled set $\mathcal{L} = \{(x^{(i)}, y^{(i)})\}_{i=1}^{|\mathcal{L}|}$ and unlabeled set $\mathcal{U} = \{x^{(i)}\}_{i=1}^{|\mathcal{U}|}$, where $x^{(i)}$s are inputs and $y^{(i)}$s are corresponding labels. At each time step $t$, a model $f$ parameterized by $\theta$ is trained on the labeled set $\mathcal{L}$ which we denoted as $f_{\mathcal{L}}$, and then a data point from the unlabeled set $\mathcal{U}$ is selected to be labeled, according to an acquisition function $A$ measuring the "information gain" of a potential data point $x$:

$$x^* = \underset{x \in \mathcal{U}}{\operatorname{argmax}} A(x, f_{\mathcal{L}}, \mathcal{L}, \mathcal{U}). \tag{1}$$

One of the most common choices of the acquisition function is *maximum entropy*, which computes the estimated entropy of the unknown label of $x$, denoted as $Y$: $A_{entropy}(x, f_{\mathcal{L}}, \mathcal{L}, \mathcal{U}) := \mathcal{H}(Y \mid x; f_{\mathcal{L}})$. Although the maximum entropy acquisition function often works well in practice, it does not consider how the model would change with a new queried data point, and so can overly prioritize inherently difficult, uninformative points (*e.g.* outliers).

Expected gradient length [11] approximates the *expected model change* by how large the gradient of the loss becomes when adding that data point, $\mathbb{E}_{y \sim p_\theta(\cdot|x)} \|\nabla \ell_\theta(\mathcal{L} \cup (x, y))\|$. BADGE [4], a more recent variant, lower-bounds the gradient norm of the last layer induced by any possible label. These approaches do not consider how the change in a model actually interacts with the data distribution: large parameter changes might give relatively small changes in predictions on most data points. They also consider a single SGD update, only roughly approximating total change over training.

One approach to alleviate this limitation is *expected error reduction* [EER; 5], given by

$$A_{EER}(x, f_{\mathcal{L}}, \mathcal{L}, \mathcal{U}) := -\mathbb{E}_{y \sim p_\theta(\cdot|x)} \sum_{i=1}^{|\mathcal{U}|} \mathcal{H}(Y^{(i)} \mid x^{(i)}; f_{\mathcal{L}^+}). \tag{2}$$

Here $f_{\mathcal{L}+}$ refers to the model trained on $\mathcal{L} \cup \{(x,y)\}$, and $\mathcal{H}(Y^{(i)}|x^{(i)}; f_{\mathcal{L}+})$ is the (estimated) entropy of the unknown label of $x^{(i)}$ in $\mathcal{U}$ using that model. Maximizing $A_{EER}$ chooses the candidate point whose label has maximal mutual information to the labels of unlabeled data, $\mathcal{I}(Y; y| f_{\mathcal{L}})$. If $f_{\mathcal{L}}$ is a fairly good model and the problem is fairly difficult, the numerical value of $A_{EER}$ is likely dominated by the sum of irreducible (aleatoric) uncertanties over the data points; it thus might be prone to noise in estimation based on those large terms.

Freytag et al. [6] propose instead *expected model output change* (EMOC), based on the difference in model predictions as measured by a distance $\mathcal{D}$ (*e.g.* the Euclidean distance between probability vectors). The acquisition function is defined as

$$A_{EMOC}(x, f_{\mathcal{L}}, \mathcal{L}, \mathcal{U}) := \mathbb{E}_{y \sim p_\theta(\cdot|x)} \sum_{i=1}^{|\mathcal{U}|} \mathcal{D}(f_{\mathcal{L}}(x^{(i)}), f_{\mathcal{L}+}(x^{(i)})). \tag{3}$$

Intuitively, we might hope that the change in outputs roughly cancels out the irreducible (aleatoric) uncertainty, leaving us to focus primarily on the decrease in model (epistemic) uncertainty [12]. Our techniques also apply to EER, but EMOC-like criteria performed better in our initial exploration.

Despite their advantages, look-ahead acquisition functions have been in practical reach only with certain types of models: EER has been applied to Naïve Bayes [5] and Gaussian random fields [13], and EMOC only to Gaussian processess [14, 15]. Neural networks generally outperform those models, but obtaining $f_{\mathcal{L}+}$ with SGD training for every candidate example in $\mathcal{U}$ is impractically expensive. In this work, we propose to approximate $f_{\mathcal{L}+}$ with a more computationally efficient proxy based on neural tangent kernels, making look-ahead acquisition functions feasible on neural networks.

**Neural tangent kernels.** Over the past few years, connections between infinitely wide neural networks trained by gradient descent and kernel methods have become increasingly clear. The NNGP approximation of Matthews et al. [16], building off a line of work stemming from Neal [17], shows that if a network's parameters are initialized with an appropriate Gaussian distribution and only the last layer is trained, the resulting function agrees with a Gaussian process with the kernel $\mathcal{K}(x, x') = \mathbb{E}_\theta[f_\theta(x)f_\theta(x')]$, where the expectation is over initializations. Jacot et al. [7], building off several immediately preceding works on optimization of wide neural networks, show that infinitely wide fully-connected neural networks also follow Gaussian process behavior, with the kernel

$$\mathcal{K}(x, x') = \mathbb{E}_\theta \left[ \left\langle \frac{\partial f_\theta(x)}{\partial \theta}, \frac{\partial f_\theta(x')}{\partial \theta} \right\rangle \right] \tag{4}$$

(derivatives here denoting Jacobians with respect to the vector of all parameters $\theta$). The also show that this kernel remains constant over the course of training, so that these networks evolve like a linear model under kernel gradient descent. Among many important follow-ups, Arora et al. [18] expand the results to convolutional networks with finite but large widths, and Novak et al. [19] provide an implementation for nearly-arbitrary network structures. Yang [20] and Yang and Littwin [21] later showed that Jacot et al. [7]'s findings are architecturally universal, extending the domain from fully-connected networks to a large class of architectures including ResNets and Transformers.

Unfortunately, these infinite-width versions of networks tend not to generalize as well as finite networks trained by SGD. Hence, we still want to conduct active learning for finite neural networks, rather than for pure NTK models. The infinite-width NTK also tends to be a mediocre proxy for the behavior of training a finite neural network, and so using it as a proxy for $f_{\mathcal{L}+}$ does not tend to work as well as we might hope (*e.g.* Figure 1b). We will thus instead use a local linearized approximation of the neural network, based on the empirical neural tangent kernel, as introduced next.

## 3 Active Learning using NTKs

In this section, we first define a neural network, and the linear model which (in the infinite-width limit) agrees with training that network. We will prove that in the infinite-width limit, sequentially retraining on increasing datasets – as in the active learning process – is the same as if we had trained from scratch on the final dataset. In practical regimes, however, the infinite-width NTK does not tend to agree with finite-width SGD very closely. We thus use a local linear approximation of the network, based on the empirical NTK, to approximate retraining in our look-ahead criteria.

**Notation.** We mostly use the same notation as Jacot et al. [7]. We define $f$ as a fully-connected neural network with $L + 1$ hidden layers numbered from 0 (input) to $L$ (output), where each layer has $n_0, \ldots, n_L$ neurons. This network has number of parameters $P = \sum_{l=0}^{L-1} (n_l + 1)n_{l+1}$: each layer has a weight matrix $W^{(l)} \in \mathbb{R}^{n_l \times n_{l+1}}$ and a bias vector $b^{(l)} \in \mathbb{R}^{n_{l+1}}$. The network is defined as $f_\theta$, where $\theta$ collects all of the parameters: $\theta = \cup_{l=0}^L \theta^l$ and $\theta^l = \text{vec}(W^{(l)}, b^{(l)})$. We denote a labeled set $\mathcal{L}$ as defined in Section 2 and use $\mathcal{X} = \{x : (x, y) \in \mathcal{L}\}$ and $\mathcal{Y} = \{y : (x, y) \in \mathcal{L}\}$ to denote its inputs and labels, respectively. We write $f_{\mathcal{D}_1} \xrightarrow[t=\infty]{\mathcal{D}_1 \cup \mathcal{D}_2} f_{\mathcal{D}_1 \cup \mathcal{D}_2}$ to mean that $f_{\mathcal{D}_1 \cup \mathcal{D}_2}$ is trained using gradient descent until convergence on the dataset $\mathcal{D}_2$, starting from $f_{\mathcal{D}_1}$.

Unless otherwise specified, the loss function associated with gradient descent training is assumed to be the squared loss, $\ell(\mathcal{Y}, f(\mathcal{X})) := \frac{1}{2} \|\mathcal{Y} - f(\mathcal{X}))\|_2^2$. This allows for far more efficient usage of NTKs, but Hui and Belkin [22] demonstrate that, with the right learning rate, $L_2$ loss is just as effective as cross-entropy for many vision and natural language processing tasks.

**Neural networks in the infinite width limit.** Jacot et al. [7] show that in the infinite width limit, when the network $f$ is trained using gradient descent with the squared loss on a labeled set $\mathcal{L}$, the outputs of the network on any arbitrary point $x$ evolve as

$$f_t(x) = f_0(x) + \mathcal{K}(x, \mathcal{X})\mathcal{K}(\mathcal{X}, \mathcal{X})^{-1}(\mathcal{I} - e^{-t\mathcal{K}(\mathcal{X}, \mathcal{X})})(\mathcal{Y} - f_0(\mathcal{X})), \tag{5}$$

where $\mathcal{K}$ is the NTK of (4), which stays constant during training. Here we treat inputs $\mathcal{X}$ as a matrix and labels $\mathcal{Y}$ as a vector in corresponding order.

Consider sequentially training a network on increasing datasets $S_1 \subset S_2 \subset \cdots \subset S_C$, warm-starting each time from the result of training on the previous dataset. We build on the results of Jacot et al. [7] to show that the final network from this process is asymptotically equivalent, in the infinite-width limit, to training the same network from scratch (cold-start) on the last, biggest dataset $S_C$. This justifies our efficient approximation of the retrained network, even after we have gone through many iterations of active learning. See Appendix A for a formal statement and proof.

**Theorem 3.1** (Informal). *Let $S_1, S_2, \ldots, S_C$ be $C$ datasets such that for all $i > j$, $S_j \subset S_i$. Let $f_0$ be a randomly initialized network. Let $f_{S_{1,2,\ldots,C}}$ be the network resulting from starting at $f_0$ and training $f$ using gradient descent on datasets $S_1, \ldots, S_C$ sequentially until convergence:*

$$f_0 \xrightarrow[t=\infty]{S_1} f_{S_1} \xrightarrow[t=\infty]{S_2} f_{S_{1,2}} \cdots \xrightarrow[t=\infty]{S_C} f_{S_{1,2,\ldots,C}}.$$

*Also, let $f_{S_C}$ be $f_0 \xrightarrow[t=\infty]{S_C} f_{S_C}$. Assuming that $f$ has $L + 1$ layers such that taking $n_0, n_1, \ldots, n_L \to \infty$ sequentially, we have for any arbitrary data point $x$ that $f_{S_C}(x) = f_{S_{1,2,\ldots,C}}(x)$.*

**NTK approximations of retrained networks.** Armed with Theorem 3.1, we can now approximate the outputs of a neural network from retraining. Suppose we have a labeled set $\mathcal{L}$ and unlabeled set $\mathcal{U}$ as defined in Section 2. A neural network $f_\mathcal{L}$ (whose layers are assumed to be infinitely wide) has been trained on $\mathcal{L}$ as $f_0 \xrightarrow[t=\infty]{\mathcal{L}} f_\mathcal{L}$. We are interested in characterizing the outcome after retraining this neural network using each of the data points in $\mathcal{U}$. In other words, for each $x' \in \mathcal{U}$ with a hypothetical label $y'$ and $\mathcal{L}^+ := \mathcal{L} \cup (x', y')$, we would like to know what $f_\mathcal{L} \xrightarrow[t=\infty]{\mathcal{L}^+} f_{\mathcal{L}^+}$ looks like.

According to (5), the outputs of $f_\mathcal{L}$ for an arbitrary data point $x$ can be formulated as

$$f_\mathcal{L}(x) = f_0(x) + \mathcal{K}(x, \mathcal{X})\mathcal{K}(\mathcal{X}, \mathcal{X})^{-1}(\mathcal{Y} - f_0(\mathcal{X})). \tag{6}$$

Let $\mathcal{X}^+$ and $\mathcal{Y}^+$ be inputs and labels of $\mathcal{L}^+$. Then, based on Theorem 3.1, we can conclude that

$$f_{\mathcal{L}^+}(x) = f_0(x) + \mathcal{K}(x, \mathcal{X}^+)\mathcal{K}(\mathcal{X}^+, \mathcal{X}^+)^{-1}(\mathcal{Y}^+ - f_0(\mathcal{X}^+)). \tag{7}$$

As $\mathcal{K}$ remains constant in the infinite-width regime, most of the relevant quantities can be reused:

$$\mathcal{K}(x, \mathcal{X}^+) = \begin{bmatrix} \mathcal{K}(x, \mathcal{X}) & \mathcal{K}(x, x') \end{bmatrix}, \qquad \mathcal{K}(\mathcal{X}^+, \mathcal{X}^+) = \begin{bmatrix} \mathcal{K}(\mathcal{X}, \mathcal{X}) & \mathcal{K}(\mathcal{X}, x') \\ \mathcal{K}(x', \mathcal{X}) & \mathcal{K}(x', x') \end{bmatrix}. \tag{8}$$

**Remark 3.2.** *In the infinite width limit, we can analytically obtain the predictions of a neural network after retraining on additional data points by augmenting its corresponding NTK using (8).*

Several works have shown that while being an inspiring theoretical motivation, neural networks in the infinite width limit do not work as well as their finite-width counterparts [18, 23, 24]. Although Arora et al. [18] prove non-asymptotic bounds between these infinite width neural networks and their corresponding finite-width networks, empirical studies in [20, 25] have shown that this approximation may not be effective for practical network widths. (If it were, we wouldn't need a network at all, and would simply do all our learning with a Gaussian process based on the NTK.) Thus, we would like to be able to efficiently characterize the outputs of a finite neural network after retraining.

Lee et al. [25] showed the first-order Taylor expansion of a neural network around its initialization (a *linearized neural network*) has training dynamics converging to that of the neural network as the width grows. That is, let $f_t$ denote the network which has been trained with gradient descent for $t$ steps, and $f_t^{lin}$ the result of training the linearized network for $t$ steps. They prove that, under some regularity conditions and when the learning rate $\eta$ is less than a certain threshold,

$$\sup_{t \geq 0} \|f_t(x) - f_t^{lin}(x)\|_2 = \sup_{t \geq 0} \|\Theta_t - \Theta_0\|_F = \mathcal{O}\left(\frac{1}{\sqrt{\text{width}}}\right). \tag{9}$$

Informally, when $f$ is wide enough, its predictions can be approximated by those of the linearized network $f^{lin}$. This is attractive since $f^{lin}$ has simple training dynamics, converging to

$$f_{\mathcal{L}}^{lin}(x) = f_0(x) + \Theta_0(x, \mathcal{X})\, \Theta_0(\mathcal{X}, \mathcal{X})^{-1}(\mathcal{Y} - f_0(\mathcal{X})), \tag{10}$$

where $\Theta_0$ is the *empirical* tangent kernel of $f_0$, $\Theta_0(x, y) := \nabla_\theta f_0(x)\, \nabla_\theta f_0(y)^\top$.

In the infinite-width limit, the empirical NTK $\Theta_t$ converges to the NTK $\mathcal{K}$ almost surely, leading (10) to become equivalent to (6). In finite-width regimes, however, it is intuitive to expect that the *local* approximation may still be able to handle *local* retraining, such as $f_{\mathcal{L}} \xrightarrow[t=\infty]{\mathcal{L} \cup \{(x,y)\}} f_{\mathcal{L} \cup \{(x,y)\}}$, even when the correspondence to the infinite-width limit $\mathcal{K}$ is loose. We thus model retraining of a finite network by augmenting the empirical NTK as in (8), as justified by Theorem 3.1 and Remark 3.2. Specifically, we approximate the finite width network $f_{\mathcal{L}^+}(x)$ defined by $f_{\mathcal{L}} \xrightarrow[t=\infty]{\mathcal{L}^+} f_{\mathcal{L}^+}$ using

$$f_{\mathcal{L}^+}(x) \approx f_{\mathcal{L}^+}^{lin}(x) = f_{\mathcal{L}}(x) + \Theta_{\mathcal{L}}(x, \mathcal{X}^+)\Theta_{\mathcal{L}}(\mathcal{X}^+, \mathcal{X}^+)^{-1}\left(\mathcal{Y}^+ - f_{\mathcal{L}}(\mathcal{X}^+)\right) \tag{11}$$

where $\Theta_{\mathcal{L}}$ is the empirical NTK obtained from the parameters of $f_{\mathcal{L}}$.

**Efficient computation.** To avoid computing the full empirical NTK with the shape of $LC \times LC$ where $L$ is the size of the labeled set and $C$ is the number of classes, we use the "single-logit" approximation as explained in Wei et al. [26] (Section 2.3) in which we only compute the corresponding NTK of the first logit of the network, i.e. $\Theta(x, y) = \nabla_\theta f^{(1)}(x)^\top \nabla_\theta f^{(1)}(y) \otimes I_C$ where $f^{(1)}(x)$ refers to the first neuron of the output of $f$ on the datapoint $x$. For a kernel regression, we can take the kronecker product out, which decreases the memory and time complexity of computing the NTK by an order of $\mathcal{O}(C^2)$.

To compute (11), we must solve linear systems with $\Theta_{\mathcal{L}}(\mathcal{X}^+, \mathcal{X}^+)$. This can also be done efficiently using the block structure of $\Theta_{\mathcal{L}}(\mathcal{X}^+, \mathcal{X}^+)$: letting $\mathbf{v} = \Theta_{\mathcal{L}}(\mathcal{X}, \mathcal{X})\Theta_{\mathcal{L}}(\mathcal{X}, x')$ and $u = \Theta_{\mathcal{L}}(x', x') - \Theta_{\mathcal{L}}(x', \mathcal{X})\Theta_{\mathcal{L}}(\mathcal{X}, \mathcal{X})^{-1}\Theta_{\mathcal{L}}(\mathcal{X}, x')$,

$$f_{\mathcal{L}^+}^{lin}(x) = f_{\mathcal{L}}^{lin}(x) + \frac{1}{u}\left(\Theta_{\mathcal{L}}(x, \mathcal{X})\mathbf{v} - \Theta_{\mathcal{L}}(x, x')\right)\left(\mathbf{v}^T(\mathcal{Y} - f_{\mathcal{L}}(\mathcal{X})) - (y' - f_{\mathcal{L}}(x'))\right). \tag{12}$$

Thus, rather than inverting $\Theta_{\mathcal{L}}(\mathcal{X}^+, \mathcal{X}^+)$ for each $x \in \mathcal{U}$, we can invert $\Theta_{\mathcal{L}}(\mathcal{X}, \mathcal{X})$ only once, then use some matrix multiplications to find $f_{\mathcal{L}^+}^{lin}(x)$ for each query point.

**Time complexity.** We now provide the time complexity of the NTK approximation proposed in Equation (11) (using efficient block computation in Equation (12)). Let $L$ be the size of the labeled set, $U$ of the unlabeled set, $P$ the number of model parameters, and $E$ the number of training epochs. The computation of $\Theta_{\mathcal{L}}(\mathcal{X}_{\mathcal{U}}, \mathcal{X}_{\mathcal{L}})$, $\Theta_{\mathcal{L}}(\mathcal{X}_{\mathcal{L}}, \mathcal{X}_{\mathcal{L}})$, and $\Theta_{\mathcal{L}}(\mathcal{X}_{\mathcal{L}}, \mathcal{X}_{\mathcal{L}})^{-1}$ take $\mathcal{O}(LUP)$, $\mathcal{O}(L^2P)$ and $\mathcal{O}(L^3)$ time, respectively. If we compute (12) for all $x \in \mathcal{U}$, the matrix multiplication takes $\mathcal{O}(UL^2)$. Putting altogether, the time complexity of the NTK approximation is $\mathcal{O}(LUP + L^2P + L^3 + UL^2)$. Since we assume $P > \max(L, U)$ for overparameterized models, the time complexity can be simplified to $\mathcal{O}(LUP + L^2P)$.

**Algorithm 1:** Active learning using NTKs

---

**Input:** Initialize a model $f_0$, labeled pool $\mathcal{L}_0$, and unlabeled pool $\mathcal{U}_0$

Train $f_0$ on the initial labeled set $\mathcal{L}_0$, using SGD, to obtain $f_{\mathcal{L}_0}$

**for** $t = 0$ **to** $T - 1$ **do**

    Compute $\Theta_{\mathcal{L}_t}(\mathcal{X}_{\mathcal{U}_t}, \mathcal{X}_{\mathcal{L}_t})$, $\Theta_{\mathcal{L}_t}(\mathcal{X}_{\mathcal{L}_t}, \mathcal{X}_{\mathcal{L}_t})$ and $\Theta_{\mathcal{L}_t}(\mathcal{X}_{\mathcal{L}_t}, \mathcal{X}_{\mathcal{L}_t})^{-1}$

    **for** $x'$ **in** $\mathcal{U}_t$ **do**

        Estimate the label for $x'$ as $y' = \operatorname{argmax} f_{\mathcal{L}_t}(x')$

        Approximate $f_{\mathcal{L}_t \cup (x', y')}$ using (11) and (12) for faster computation

        Track $x^*$ as the point with maximal $A_{MLMOC}(x', f_{\mathcal{L}_t}, \mathcal{L}_t, \mathcal{U}_t)$ we've seen so far

    **end for**

    Obtain the label $y^*$ for $x^*$ from the oracle

    Update the labeled set $\mathcal{L}_{t+1} = \mathcal{L}_t \cup \{(x^*, y^*)\}$ and unlabeled set $\mathcal{U}_{t+1} = \mathcal{U}_t \setminus \{(x^*, y^*)\}$

    Train $f_{\mathcal{L}_t}$ on $\mathcal{L}_{t+1}$, using SGD, to obtain $f_{\mathcal{L}_{t+1}}$

**end for**

---

The dominant term for naïve SGD retraining is $\mathcal{O}(LUPE)$ whereas the proposed method using block structure is dominated by $\mathcal{O}(LUP)$ time if $U > L$, otherwise $\mathcal{O}(L^2 P)$ time. If $U > L$, then the proposed method is faster by an order of $\mathcal{O}(E)$. In the other case, if $L > EU$, one can use the conjugate gradient kernel regression solvers as in Rudi et al. [27] and Gardner et al. [28] that further reduce the time complexity of our proposed method to $\mathcal{O}(\max(PL\sqrt{L}, PU\sqrt{U}))$ depending on whether $L > U$ or $L < U$, which is faster than the naïve SGD training by an order of $\mathcal{O}(UE/\sqrt{L})$ or $\mathcal{O}(LE/\sqrt{U})$.

**MLMOC querying.** Instead of the EMOC acquisition function (3), we define a new function more suitable for a linearized network, which we term *Most Likely Model Output Change* (MLMOC):

$$A_{MLMOC}(x', f_{\mathcal{L}}, \mathcal{L}_t, \mathcal{U}_t) := \sum_{x \in \mathcal{U}} \|f_{\mathcal{L}}(x) - f_{\mathcal{L}^+}^{lin}(x))\|_2 \tag{13}$$

where $\mathcal{L}^+ = \mathcal{L} \cup \{(x', y')\}$ with $y' = \operatorname{argmax} f_{\mathcal{L}}(x')$. We only consider the most likely pseudo-label, instead of the expectation; this saves a significant amount of computation, by a factor of the number of classes, but empirically does not hurt accuracy. We also suspect that, when $f_{\mathcal{L}}$ is already reasonably accurate, we can "trust" the most likely label more than we can trust accurate estimation of probabilities for low-probability losses, especially when training networks with $L_2$ loss. Algorithm 1 shows pseudo-code for the proposed algorithm.

**Sequential query strategy.** In the active learning literature, each query of new data points is essentially always followed by retraining the underlying model. This, however, may not be practical for the cases where annotation is quick but SGD training of a large neural network is slow. Although there has been significant effort towards batch selection strategies based on finding diverse examples to query in a batch [e.g. 4, 29, 30], which among other benefits can minimize the number of times we must retrain with SGD, they cannot fully take advantage of low-latency annotations if they are available. (This may be the case if, for instance, labeling tasks can be quickly pushed out to a pool of human labelers available for many such annotation tasks at once, or other cases where a limited number of queries are desired to avoid expense, damage to a system being measured, or so on, but those individual measurements can still be taken promptly.)

Our proposed linearized network can take advantage of sequentially added annotations without requiring SGD retraining steps. Given an empirical NTK $\Theta$, adding another training point $(x, y)$ requires nothing more than adding a row and column to the kernel. As a result, computing $f^{lin}$ and following $A_{MLMOC}$ with an additional data point $(x, y)$ is almost instant. This approximation can utilize the information from the label $y$ directly while batch selection strategies cannot. (After adding a substantial number of points, we will want to retrain, but it is not needed after every new data point.) We experimentally demonstrates the effectiveness of this sequential query strategy in Section 5.

# 4 Related Work

Two main approaches in pool-based active learning are uncertainty and representation-based methods. The general goal of the uncertainty-based methods is to query the most "informative" data points, thus, their acquisition functions vary depending on how to measure the informativeness of a data point. As simple but effective uncertainty-based querying methods, posterior probability [31, 32], entropy [1], margin sampling [33] and least confident [11], have been widely used in practice. They do not, however, take into account how a candidate data point would change a model, or how this change would interact with unseen examples. More advanced querying methods have thus been proposed as discussed in Section 2. There are also Bayesian uncertainty-based methods, particularly Bayesian Active Learning by Disagreement (BALD) [2], which measures the mutual information between a new data point and model parameters, which Gal et al. [34] propose to estimate efficiently with Monte Carlo dropout networks [35].

The goal of the representation-based methods is to query examples that are the most "representative" among the unlabeled set $\mathcal{U}$, hoping that doing well on those examples leads to doing well on the whole unseen dataset. Sener and Savarese [36] define a core set as a batch of images that minimize the distance between unlabeled and labeled images if added to the labeled pool. Kirsch et al. [30] further expand BALD to a batch setting using a greedy algorithm. Bıyık et al. [29] propose to use determinantal point processes to target diversity of the queried examples while maintaining informativeness. BADGE [4] measures the uncertainty using a gradient norm but also encourage the diversity of queried images using k-means++ seeding [37].

Other approaches include that of Yoo and Kweon [38], who employ an additional loss prediction module to an existing neural network. However, the additional loss computation module can make training unstable (as we observed in some experiments). Tran et al. [39] improves BALD by adding a variational autoencoder and auxiliary classifier generative adversarial network, from which a synthesized data point $x'$ similar to the queried image $x^*$ is generated.

Our proposed method is different from the previous works in that it "looks one step ahead" instead of relying on the current state of a model. Perhaps the most similar proposal to ours is that of Borsos et al. [40], who propose using the infinite NTK to solve a bi-level optimization problem for semi-supervised batch active learning. Aside from the difference in our objective functions, however, using the infinite NTK to approximate a neural network in active learning does not seem to be a good choice; as the size of training set increases, the error of the approximation can be quite large. We explore the difference between infinite and empirical NTKs experimentally in Section 5.

# 5 Experiment Results

**Datasets.** To demonstrate the effectiveness of the proposed method, we provide experimental results on three benchmark datasets for classification tasks: MNIST [8], SVHN [9], CIFAR10 [10], and CIFAR100 [10]. MNIST consists of 10 hand-written digits with $60,000$ training and $10,000$ test images, at size $28 \times 28$. SVHN also consists of 10 digit numbers with $73,257$ training and $26,032$ test images, at size $32 \times 32$; it is more challenging than MNIST, containing images from pictures of house numbers, with much more variation and many distractions present. CIFAR10 contains $50,000$ training and $10,000$ test images, also $32 \times 32$ and equally split between 10 classes like `airplane`, `frog`, and `truck`. CIFAR100 is the same as the CIFAR10, except it has 100 classes; the resolution is still $32 \times 32$, and images are equally split between 100 classes. The 100 classes are grouped with 20 super-classes, but we use only the 100 sub-classes in this work.

**Implementation.** We implement a pipeline for the proposed method using PyTorch [41] and Jax [42]; our neural networks $f$ are implemented in PyTorch, whereas the linearized models $f^{lin}$ are implemented in Jax with the `neural-tangents` library [43]. We employ a ResNet18 [44] and WideResNet [45] with one or two layers and maximum width of $640$, which is wide enough for a linearized neural network to be a good approximation, while still being powerful enough for these datasets. As we will see in our experiments, even for narrower networks, the proposed method outperforms random acquisition strategy by large margins.

As mentioned earlier, we use $L_2$ loss, rather than cross-entropy; this allows for a faster NTK approximation, as in the formulas of Section 3, rather than requiring differential equation solvers.

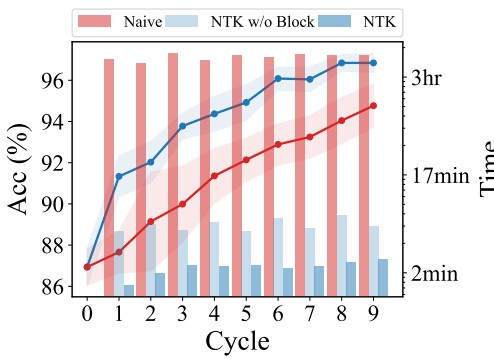

(a) Naïve look-ahead acquisition versus NTK approximation. Bars show runtime per cycle.

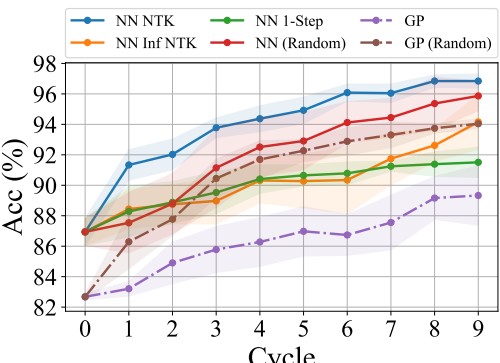

(b) MLMOC on various models, along with random-acquisition baselines.

Figure 1: Comparisons of several related approaches on MNIST.

| Methods | NN NTK | NN 1-step | NN Inf NTK | GP |
|---|---|---|---|---|
| NN NTK | 1.0000 | 0.0109 | 0.0093 | 0.0014 |
| NN 1-step | 0.0109 | 1.0000 | 0.6863 | 0.0422 |
| NN Inf NTK | 0.0093 | 0.6863 | 1.0000 | 0.0742 |
| GP | 0.0014 | 0.0422 | 0.0742 | 1.0000 |

Table 1: The p-values of the post-hoc paired t-test.

Appropriately tuned $L_2$ loss is as effective as cross-entropy [22]. To ensure that the empirical NTK of the training data ($\mathcal{X}_\mathcal{L}$) is positive semidefinite, in batch normalization layers, we freeze the current running statistics. We implement LL4AL [38] and BADGE [4] based on their publicly available code. We provide anonymous code for the full pipeline in the supplementary material.

**Query scheme.** Following active learning literature, for each query step, we randomly draw a subset from an unlabeled set, and query a fixed number of examples from the subset. To build a batch, we take the points with the highest $A_{MLMOC}$ scores, which we found to perform well without any batch diversity requirements. At each cycle, we query 20, 1 000, 1 000, and 1 000 new data points on MNIST, SVHN, CIFAR10, and CIFAR100 from the subset of 4 000, 6 000, 6 000, and 6 000 unlabeled data points, and initialize the labeled set $\mathcal{L}$ with 100, 1 000, 1 000, and 10 000 data points.

**Making look-ahead acquisitions feasible.** In Figure 1a, we demonstrate the NTK approximation indeed makes it feasible to use look-ahead acquisition functions. We run the MLMOC acquisition function with naïve SGD retraining (Naive) and proposed NTK approximation with (NTK) and without (NTK w/o Block) block computation in (12) on MNIST. For the naïve version, we retrain the neural network for 15 epochs using SGD. The line plots represent accuracy whereas bar plots represent wall-clock time to query data, using a NVIDIA V100 GPU. The NTK approximation is not only computationally efficient (more than 100 times faster than Naïve) but also yields significantly more accurate models. Although one might expect that Naïve is an upper bound for the NTK approximation, as it actually "looks ahead," in reality it is hard to retrain a neural network until convergence for every candidate data point; a "proper" look-ahead method should perhaps look at an ensemble of several training runs, with different learning rate schedules, etc.

**Comparison of look-ahead acquisitions.** Having seen that the empirical NTK effectively approximates (even outperforms) the Naïve look-ahead acquisition function, we now show the superiority of the empirical NTK over existing look-ahead methods. In Figure 1b, we provide four look-ahead methods using MLMOC acquisition function along with their corresponding random baselines (Random). NN refers to a neural network-based model whereas GP refers to a model based on Gaussian process. Instead of using empirical NTK as we propose (denoted as NN NTK, blue), Borsos et al.

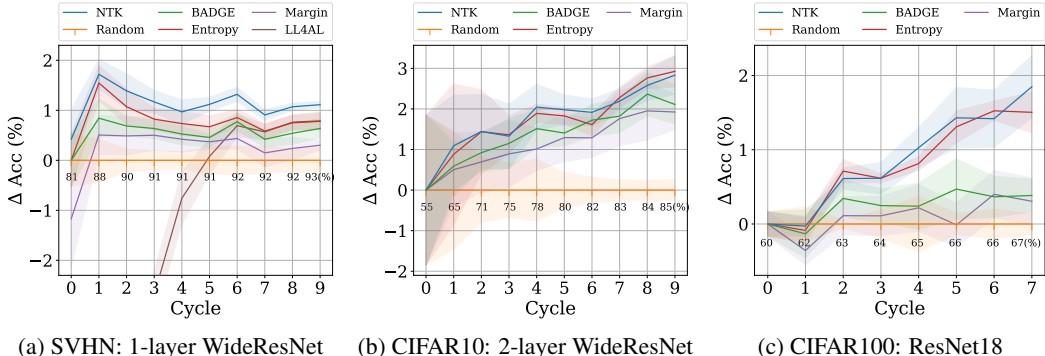

(a) SVHN: 1-layer WideResNet     (b) CIFAR10: 2-layer WideResNet     (c) CIFAR100: ResNet18

Figure 2: Comparison of the-state-of-the-art active learning methods on various benchmark datasets. Vertical axis shows difference from random acquisition, whose accuracy is shown in text.

[40] use infinite NTK (NN Inf NTK, orange). Käding et al. [14] approximate SGD retraining using only one-step SGD update (NN 1-step, green) whereas Käding et al. [15] exactly compute look-ahead acquisition using GP for regression problems. We modify their model for classification tasks, use an infinite NTK, and denote it as GP (purple). As shown in Figure 1b, ours is the only of these methods to outperform random baselines here. Neither infinite NTK nor 1-step approximation are good approximation for retraining steps, especially as more data points are added. Also, although GP [15] works well on regression tasks, it struggles on classification tasks.

To validate the difference between different look-ahead strategies, we conduct Friedman's test followed by post-hoc paired t-test on the existing look-ahead strategies in Figure 1b: NN NTK, NN 1-step, NN Inf NTK, and GP. Friedman's test is a non-parametric statistical test. The null hypothesis in our case is that the population mean of the performance of each look-ahead strategy is the same and the alternative hypothesis is that at least one population mean is different from the rest of look-ahead strategies. The p-value of the Friedman's test is $0.0045$ ($< 0.05$). Hence, we reject the null hypothesis. As it means at least one population mean is different from the rest of look-ahead strategies, we further conduct the post-hoc paired t-test to check where the significant difference is coming from. The result of the post-hoc paired t-test is shown in Table 1. Here, the null hypothesis for each pair is that the population mean of two methods is the same. As all the p-values between ours (NN NTK) and the other methods are less than the conventional significance level ($0.05$), we conclude that our proposed method is significantly better than the other look-ahead strategies.

**Comparison with state-of-the-art.** In Figure 2, we compare the proposed NTK active learning to state-of-the-art methods – Random, Entropy, Margin sampling (Margin) [46], BADGE [4], and LL4AL [38] – on SVHN, CIFAR10, and CIFAR100 with different architectures; we provide additional results in Appendix B. For clarity, we show the difference between each method and Random. The numbers below Random line give the accuracies of Random at each cycle. As mentioned earlier, training LL4AL is often unstable due to loss computation module; in particular, it performs too poorly to show on the same plot for Figure 2b and Figure 2c. We show the instability of training LL4AL with varying learning rate in Appendix C. Each experiment is run for six different seeds; lines show the mean performance, and shading shows a $95\%$ confidence interval for the mean. Figure 2(a) shows that the NTK method outperforms all the state-of-the-art methods throughout training on SVHN. A similar pattern is observed on CIFAR10 and CIFAR100; NTK is consistently comparable to or better than the best competitor methods, far outperforming existing look-ahead approaches.

**Sequential query strategy.** As mentioned in Section 3, one great advantage of the NTK approximation is that it can exploit additional annotations without needing to run SGD training for a neural network, which cannot be done in existing active learning methods for neural networks. On MNIST, Figure 3a compares sequential NTK where the oracle provides true label for every new queried data point, to batch NTK where annotation is done batch-wise as with a normal active learning setting. Sequential NTK outperforms batch NTK; especially, sequential NTK reaches more than $94\%$ in one cycle, and in cycle 3, it has already reached $96\%$ accuracy, where batch NTK plateaus. We expect the sequential NTK can be useful in practice when annotation is much faster than SGD training.

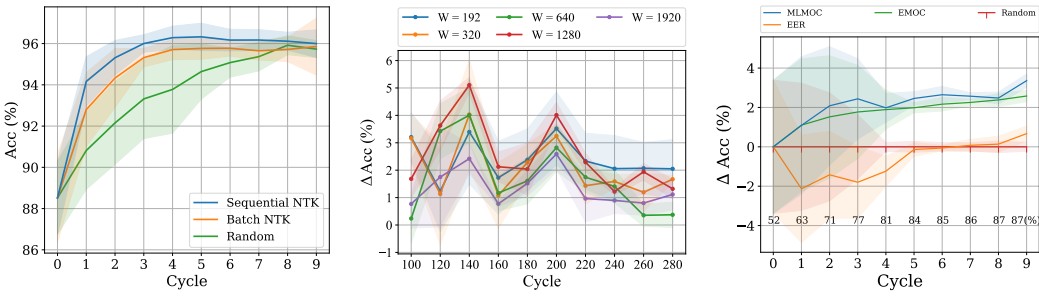

| (a) Sequential *vs.* batch querying. | (b) Performance with varying width. | (c) Diff. look-ahead strategies. |

Figure 3: Further comparisons of algorithm variants on MNIST.

**Varying model widths.** As shown in Equation (9), the NTK approximation gets closer to the dynamics of a neural network as the width of the neural network increases. But, since very wide networks are computationally expensive in practice, it is important to empirically demonstrate that the NTK approximation works even with reasonably narrow networks. To this end, we vary the maximum width of the WideResNet with one block layer we use for the experiments as shown in Figure 3b. We can observe that regardless of the maximum width, the NTK method significantly outperforms Random strategy until convergence, indicating that the NTK approximation is good enough for the purpose of active learning.

**Comparison of different acquisition strategies.** We compare different look-ahead acquisition functions in Figure 3c. As mentioned with Equation (13) in Section 3, MLMOC and EMOC are not significantly different. As MLMOC is computationally cheaper than EMOC proportional to the number of classes, we use MLMOC over EMOC as an acquisition function. But note that, the proposed NTK approximation is applicable to any look-ahead acquisition functions including MLMOC, EMOC and EER. Thus, we hope that our method would provide a framework for other researchers to build upon and propose various look-ahead active learning methods or enhance the look-ahead approximation to help further tackle the problem of pool-based active learning.

## 6 Conclusion

We propose an algorithm to make look-ahead acquisition strategies feasible for active learning with fairly general neural network architectures. We prove that the outputs of wide enough neural networks can be efficiently approximated using a linearized network using empirical neural tangent kernels. We empirically show that this approximation is at least 100 times faster than naïve computation of look-ahead strategies, while even being more accurate. Armed with this approximation, we significantly outperform previous look-ahead strategies, and achieve equal or better performance compared to the state-of-the-art methods on four widely used benchmark datasets in active learning regime. We also propose a new sequential active learning algorithm that decouples querying and SGD training for the first time.

One limitation of the NTK approximation is that it is still slower than common "myopic" active learning methods such as entropy and BADGE. Compared to our proposed method where it takes $\mathcal{O}(\max(PL\sqrt{L}, PU\sqrt{U}))$ using conjugate gradient kernel regression solvers, entropy requires $O(PU)$ time (using notations in Section 3). Improving time complexity of the NTK approximation would be an interesting future work along with devising better look-ahead acquisition functions that improve performance.

## Acknowledgments and Disclosure of Funding

This research was enabled in part by support, computational resources, and services provided by the Canada CIFAR AI Chairs program, the Natural Sciences and Engineering Research Council of Canada, Advanced Research Computing at the University of British Columbia, WestGrid, Calcul Québec, and the Digital Research Alliance of Canada. We would like to specifically thank Roman Baranowski for helping us with our computational requirements.

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
