# OpenReview forum: "Making Look-Ahead Active Learning Strategies Feasible with Neural Tangent Kernels"
_NeurIPS.cc/2022/Conference — NeurIPS 2022 Accept_

### Official Review · Reviewer_ywhB · 2022-07-10

**Rating:** 6
**Confidence:** 1
**Soundness:** 2 fair
**Presentation:** 2 fair
**Contribution:** 2 fair

**Summary:**

The paper proposes an active learning method for neural networks that leverages neural tangent kernels to approximate the network. The proposed approach leverages lookahead to predict the expected NN output change when it is retrained with a specific input pair. The paper further proposes an acquisition function, Most Likely Model Output Change (MLMOC). The authors compare the proposed approach with state-of-the-art methods on three benchmark datasets for classification tasks and show that it either outperforms existing methods or performs equally well.

**Questions:**

**a)** Why is your acquisition function more suitable for a linearised network?

**b)** In algorithm 1 (you should add line numbers) you mention an oracle, but I don't see any mention to it anywhere else. Where does the oracle come from?

**c)** You mention lookahead but it is not clear to me if it is lookahead of: a) 1 point/label; b) n points/labels; c) 1 batch; d) n batches?

**d)** What is the meaning of T in algorithm 1?

**e)** Are there non-NN active learning baselines? How would your method fare against them?

**f)** Why do you only use MLMOC in Figure 1.b)? Why not test also with different acquisition functions? Also, in this experiment, why do you modify the models to "use an infinite NTK" (line 305-206) when you know that infinite NTK perform worse? You should aim to beat the strongest baselines, not the weakest.

**Comments:**
- Line 104: typo "the" also show => "they" also show

**Limitations:**

Authors address some limitations of their work (time complexity). Is this method only applicable to classification tasks? How does it generalize to regression tasks?

**Strengths And Weaknesses:**

**Originality:** Related work seems to be adequately cited and the differences between the paper and previous work are explained.

**Quality:** The claims are somewhat well supported. For example, in line 210 it is stated that "empirically does not hurt accuracy" however I did not see a comparison between the acquisition functions.

**Clarity:** The paper makes a good job at introducing the active learning setting however it then starts getting confusing, particularly the introduction to the lookahead setting lacks details. For instance, in Section 3, the contribution and insights of the paper are too diluted with existing work. Specifically, having subsections would be useful to distinguish the different steps more clearly: 1) sequential training of the network (theorem 3.1); 2) approximating the retrained network. I felt that the part on "NTK approximations of retrained networks" after remark 3.2 was diluting your conclusions with those of related work. Here perhaps you could start by discussing the limitations/conclusions of related work, explaining how you build on them.

**Significance:** It is likely that other researchers will use the ideas presented in the paper or build on them.

---

> ### Author Response · Authors · 2022-08-02
> **Response to the Reviewer ywhB (1/2)**
>
> Thank you for your comments, which will improve our paper. We respond to various specific questions below; we’ll integrate these responses into the paper in revision. If anything remains unclear or you think it needs further discussion, please do continue the conversation!
>
> > 1. The claims are somewhat well supported. For example, in line 210 it is stated that "empirically does not hurt accuracy" however I did not see a comparison between the acquisition functions.
>
> Thanks for noting that we should include these experiments. We‘ve added a comparison of MLMOC and EMOC in our revised Appendix G. As claimed in line 210, there is no significant difference between MLMOC and EMOC in terms of accuracy, but MLMOC is faster by a factor of the number of classes.
>
> > 2. The paper makes a good job at introducing the active learning setting however it then starts getting confusing, particularly the introduction to the lookahead setting lacks details.
>
> Thank you for your suggestions here; we included the related work here to justify the approximation once it’s been introduced, but we will add the subsection headers you suggest and more explicit separation of related work with paragraph titles.
>
> > 3. Why is your acquisition function more suitable for a linearised network?
>
> We don’t claim that MLMOC is more suitable for a linearized network in particular; rather, the linearized network works for any look-ahead acquisition function, including MLMOC, EMOC, and EER. We chose MLMOC for additional computation savings, by a factor of the number of classes, without hurting accuracy (as discussed in line 210 and the new Appendix G).
>
> > 4. In algorithm 1 (you should add line numbers) you mention an oracle, but I don't see any mention to it anywhere else. Where does the oracle come from?
>
> We’ll add line numbers; thanks for the suggestion. The term “oracle” is widely used in the active learning literature to refer to an annotator for a given input, e.g. an image; it could be a human annotator, the result of an autonomous robotic agent, or any other source of data.
>
> > 5. You mention lookahead but it is not clear to me if it is lookahead of: a) 1 point/label; b) n points/labels; c) 1 batch; d) n batches?
>
> As mentioned in Line 34-35, we refer to “acquisition strategies which consider full retraining of a model based on hypothetical observations as look-ahead criteria.” Here, hypothetical observations refer to 1 or more points with its hypothetical labels (not true labels). For MLMOC acquisition, we obtain a hypothetical label by taking the top-scoring class labels. We look ahead 1 point at a time, since looking at $n$ data points is computationally infeasible: there are $U$ choose $n$ cases, where $U$ is the number of points in the unlabeled set.
>
>
> > 6. What is the meaning of T in algorithm 1?
>
> $T$ refers to the number of cycles; in other words, the number of times that we add new labeled data points. We’ll clarify this in revision.
>
> > 7. Are there non-NN active learning baselines? How would your method fare against them?
>
> We do provide the results of Gaussian process-based methods (GP) in Figure 1b (purple line). Note that, although this is based on [16], they focus on regression; we had to modify the method slightly for multiclass classification. We also changed their method to an infinite NTK rather than the RBF kernel based on SIFT features that they used; see the response to reviewer nYBt for a comparison to that kernel. These methods perform poorly compared to neural network-based active learning methods.
>
> We also mentioned in Section 2 an algorithm based on EER with Naive Bayes. Since Naive Bayes is unlikely to work on images, we did not consider it a comparable baseline to run here.

---

> ### Author Response · Authors · 2022-08-02
> **Response to the Reviewer ywhB (2/2)**
>
> > 8. Why do you only use MLMOC in Figure 1.b)? Why not test also with different acquisition functions? Also, in this experiment, why do you modify the models to "use an infinite NTK" (line 305-206) when you know that infinite NTK perform worse? You should aim to beat the strongest baselines, not the weakest.
>
> We observed that there is no significant difference between MLMOC and EMOC in terms of performance, as mentioned in in question 1, but MLMOC saves computation. We’ve added additional results with EMOC and other related functions in Appendix G, as mentioned.
>
> The infinite neural tangent kernel (NTK) is worse than a regular neural network or its corresponding empirical NTK because there is no feature learning. However, it is generally considered to be among the best general-purpose kernels known so far, typically far outperforming default kernels in practice. Since GP-based methods require a single kernel fixed a priori, the infinite NTK is among the best known choices for a GP.
>
> To make the difference between NN NTK, NN Inf NTK and GP in Figure 1b clear, let’s divide the models into two parts: a model trained on the labeled set, and a model to query new examples. NN NTK uses a regular NN for the training model, and its linearized approximation for the query model. NN Inf NTK uses a regular NN for the training model and infinite NTK for the query model. Lastly, GP uses the same infinite NTK for both training and query models.
>
> > 9. Authors address some limitations of their work (time complexity). Is this method only applicable to classification tasks? How does it generalize to regression tasks?
>
> The method is readily applicable to regression tasks, or indeed many other tasks since we can linearize a network with any output dimension. For this work, however, we focused on the standard active learning setting of multiclass classification to easily compare with existing methods; there is comparatively little work on active regression.
>
> > 10. Line 104: typo "the" also show => "they" also show
>
> Thanks; we’ll fix.

---

> ### Comment · Reviewer_ywhB · 2022-08-05
> **Thank you for answering the questions**
>
> I would like to thank the authors for their clarifications. I have no further questions.

---

### Official Review · Reviewer_gpbw · 2022-07-11

**Rating:** 5
**Confidence:** 4
**Soundness:** 3 good
**Presentation:** 3 good
**Contribution:** 2 fair

**Summary:**

This work proposes a new method for approximating active learning acquisition strategies. Specifically, it uses the neural tangent kernel to approximate the result of retraining, namely it linearly approximates the network and mimics the new output of adding a new sample. Then it compares the network output without training on the sample (x) and the linearly approximated network with training on (x), and accordingly selects (x).

**Questions:**

Please see the above questions.

**Limitations:**

 the authors did not well discuss the limitations, and did not discuss potential negative societal impact of their work.

**Strengths And Weaknesses:**

Pros:
Using the neural tangent kernel to approximate the result of retraining greatly reduces the training cost and improves the efficiency of active learning, compared with the previous one.

The authors provide some experimental results to support its effectiveness.


Cons:
Using the neural tangent kernel method to approximate a network is actually very hard. This is also mentioned by the authors. Although the authors use a local NTK which may improve the accuracy, I still think there is a big gap between NTK network and the vanilla one, since the width is not huge in practice. So how do the authors guarantee the quality of this local NTK? For example, do the authors have some theory to guarantee or have compared vanilla network output and this new NTK network output?

Efficiency: in algorithm 1, for each epoch, one needs to pass the whole unlabelled dataset to select proper samples. For each sample selection, one further needs to use eqn (12) to compute the NTK output. But when the lablled dataset is large, e.g. imagenet 1K, NTK kernel Theta would be very large. So computing its inverse may be expensive. In this way, it seems it is hard to scale this method to the large-scale dataset.

This labeled data plus unlabelled data is actually a semi-supervised setting. So it is necessary to compare with some classical semi-supervised methods, e.g. fixmatch, paws, etc. It is very valuable to see 1 ) whether there is performance improvement or drop compared with classical semi-supervised methods; 2) whether the proposed active learning can greatly improve the training efficiency of classical semi-supervised methods. For performance, it seems the results on CIFAR10 reported in this work is much lower than the vanilla fixmatch method.

---

> ### Author Response · Authors · 2022-08-02
> **Response to the Reviewer gpbw (1/2)**
>
> Thank you for your comments, which will improve our paper. We respond to various specific questions below; we’ll integrate these responses into the paper in revision. If anything remains unclear or you think it needs further discussion, please do continue the conversation!
>
> > 1. Using the neural tangent kernel method to approximate a network is actually very hard. This is also mentioned by the authors. Although the authors use a local NTK which may improve the accuracy, I still think there is a big gap between NTK network and the vanilla one, since the width is not huge in practice. So how do the authors guarantee the quality of this local NTK? For example, do the authors have some theory to guarantee or have compared vanilla network output and this new NTK network output?
>
> We understand this concern, and thank the reviewer for their attention here. Please see our separate comment about the approximation quality, since another reviewer also asked about this.
>
>
> > 2. Efficiency: in algorithm 1, for each epoch, one needs to pass the whole unlabelled dataset to select proper samples. For each sample selection, one further needs to use eqn (12) to compute the NTK output. But when the lablled dataset is large, e.g. imagenet 1K, NTK kernel Theta would be very large. So computing its inverse may be expensive. In this way, it seems it is hard to scale this method to the large-scale dataset.
>
> We agree that naive solution of linear systems with the NTK kernel is very expensive when the number of labeled points is large ($\\mathcal O(L^3)$ for $L$ labeled points), which would be a significant problem with very large labeled sets. However, since our acquisition strategy eventually boils down to kernel regression as shown in (11) and (12), we can use the broad literature of efficient computation of predictive means of Gaussian Processes (GPs) to efficiently compute our acquisition function more effectively. For instance, instead of Cholesky decompositions, one can use conjugate gradient descent to compute ${\Theta_{\mathcal{L}}(\\mathcal{X}^+,  \\mathcal{X}^+)}^{-1} (\\mathcal{Y}^+ - f_{\\mathcal{L}}(\\mathcal{X}^+))$ in $\\mathcal O(kL^2)$ time, where $k$ is the number of gradient updates; in practice, $k<100$ usually suffices (Golub et al., “Matrix computations”). As mentioned in Section 3 and Appendix B, this can be further reduced to down to $\\mathcal O(L\sqrt{L})$ using FALKON [27], also based on conjugate gradient. Inducing point or other approximations can also find approximate solutions in time linear in $L$.
>
> We did not do this in this work, because it was not necessary for experimentation on the datasets we tried (where $L$ was never very large – and indeed, active learning is most useful when $L$ is small), but these approaches should be possible to incorporate with no changes to our active learning algorithm.
>
>
> > 3. This labeled data plus unlabelled data is actually a semi-supervised setting. So it is necessary to compare with some classical semi-supervised methods, e.g. fixmatch, paws, etc. It is very valuable to see 1 ) whether there is performance improvement or drop compared with classical semi-supervised methods; 2) whether the proposed active learning can greatly improve the training efficiency of classical semi-supervised methods. For performance, it seems the results on CIFAR10 reported in this work is much lower than the vanilla fixmatch method.
>
> There is indeed a line of semi-supervised active learning methods, such as Borsos et al. [9] and Gao et al (“Consistency-based Semi-supervised Active Learning: Towards Minimizing Labeling Cost”), that use the unlabeled set for semi-supervised learning and then perform active learning on the resulting model. However, most active learning works, like LL4AL, BADGE, or the ones mentioned above or in Zhan et al. (“A Comparative Survey of Deep Active Learning”) do not consider using the unlabeled set in this way. Although it should be possible to use our approach in this active semi-supervised setting with little or no modification to the technique, and this would be a very reasonable thing to do, we wanted to stick to the more common setting for comparison here.
>
> Also, in most recent pool-based active learning work (including ours as well as LL4AL, BADGE, etc), we do not necessarily have access to all of the unlabeled data, but instead a small pool of candidate points which gets updated at every iteration (cycle) of the algorithm. In this setting, the semi-supervised pool is not so big, and so probably does not help as much as it might in standard semi-supervised settings (where the amount of unlabeled data is far larger than the amount of labeled data).

---

> ### Author Response · Authors · 2022-08-02
> **Response to the Reviewer gpbw (2/2)**
>
> > 4. The authors did not well discuss the limitations, and did not discuss potential negative societal impact of their work.
>
> For limitations: we did discuss the added computational complexity of our work compared to standard “myopic” methods in Section 3 and Appendix B, including a detailed explanation of the time complexity; we will add a more explicit reference to this e.g. in the conclusion, and will add some more discussion of this. Outside of this, we’re not aware of other significant limitations; if you have any others to suggest, we’d be happy to hear them.
>
> We are not aware of any particular potential negative impact of our work compared to other pool-based active learning approaches, a well-established sub-field of machine learning research that doesn’t seem particularly harmful in itself, though we will add a mention of being careful about potential misuse of active learning relative to human decision-making. We would be happy to hear any potential negative impact of our work in particular, or any work to cite on problems with active learning in general, that we might have missed.

---

> ### Author Response · Authors · 2022-08-08
> **Discussion ends in less than 2 days**
>
> Dear reviewer,
>
> We would like to inform you that we have added a section on theoretical analysis of our approximation quality in the revised version (**Appendix K**).
>
> Since you mentioned that
>
> > I still think there is a big gap between NTK network and the vanilla one, since the width is not huge in practice. So how do the authors guarantee the quality of this local NTK? For example, do the authors have some theory to guarantee or have compared vanilla network output and this new NTK network output?
>
> we wanted to inquire whether you were satisfied with our responses below, and our newly added sections in the appendix (**Appendix H**, discussing the experimental verification of our approximation quality and **Appendix K** regarding the theoretical analysis of the approximation quality for retraining). If not, please let us know soon so that we could provide more details or address your other questions/concerns before the discussion period ends.
>
> Thank you for your time and attention!

---

> > ### Comment · Reviewer_gpbw · 2022-08-09
> > **thanks for your response**
> >
> > Thanks for your detailed response. Except the approximation quality and efficiency, my other concerns are well addressed.
> >
> > For approximation quality, I think there are two issues that should be solved. 1）For the experiments, the authors only use a one-layered network to verify it.  This may be not the real case, since in practice one often use deep network, e.g. ResNet50 or Transformer (e.g. 12 layers). For a small network, the results that linear approximation can give a good approximation is expected and cannot well verify youthe r point. For theory part, the authors also give the approximation for one layer. Though for one layer, using its linear approximation can have a bound depending on the inverse of width. However, the bound could exponentially increase along with depth, right? So in my view, the authors should consider the bound of approximation of multi-layers which is the case in this work.
> >
> > For efficiency, the authors claim there are other methods that can boost efficiency. But the issue is that the authors do not provide any empirical support, e.g. experimental results. So this claim is not very convincing.
> >
> > Based on these factors, I do not change my score now.

---

> > > ### Author Response · Authors · 2022-08-09
> > > **Thanks for your response**
> > >
> > > Dear reviewer,
> > >
> > > We would like to thank you for your response and further mentioning your concerns and questions. We would also like to comment on your concerns in this comment:
> > >
> > > > 1）For the experiments, the authors only use a one-layered network to verify it. This may be not the real case, since in practice one often use deep network, e.g. ResNet50 or Transformer (e.g. 12 layers). For a small network, the results that linear approximation can give a good approximation is expected and cannot well verify youthe r point.
> > >
> > > We would like to clarify that the network that we used in the experiments (which we call 1- (block) layer WideResNet) in fact has **18 PyTorch layers**, and its maximum width is 160 (channels for convolution layer), which is rather small (compared to ResNet18 per say, that has a maximum width of 512). The source code of this model [can be found here](https://github.com/anonymous-ml-enjoyer/ll4al-reproduce-lr/blob/c631debaa4209d525b1a08580bdc19a8eb4df5ef/models/wide_resnet.py#L138), and is also available in the provided anonymous source code in the supplementary materials. It is based on the model WRN-28-1 [45]. We will clarify this in the revision to avoid this confusion. If time allows us, we will get back to you with the repeated experiment on deeper models, but we would like to also mention that to the best of our knowledge, most recent works in deep active learning have used ResNet18 as their backbone architecture, which is among the models that we have used in our experimental evaluations.
> > >
> > > > For theory part, the authors also give the approximation for one layer. Though for one layer, using its linear approximation can have a bound depending on the inverse of width. However, the bound could exponentially increase along with depth, right? So in my view, the authors should consider the bound of approximation of multi-layers which is the case in this work.
> > >
> > > Thanks for the question. We would like to clarify that in the theory part, we do not have any assumption regarding the depth of the network. As we have mentioned in the appendix, in section K we follow the same notation and proof technique of (Lee et al. 2019 - [25]), and consider the case where all of the layers in the network have equal width. As shown in section G of [25], this result expands to the case where the width of different layers follow $\frac{n_l}{n_{l'}} \to \alpha_{l,l'} \in (0, \infty)$. Moreover, this result does not depend on the depth of the network at all, and with the given $\lambda^*$, the approximation error can be upper-bounded by a constant times inverse of square root of width. We will also clarify this in the revised version to avoid any confusion.
> > >
> > > > For efficiency, the authors claim there are other methods that can boost efficiency. But the issue is that the authors do not provide any empirical support, e.g. experimental results. So this claim is not very convincing.
> > >
> > > We would like to mention that the methods that we mentioned [are currently being used in practice](https://github.com/cornellius-gp/gpytorch), but we were able to run our experiments (i.e., CIFAR100 experiments) without needing them as in our case $L$ is rather small, and the mentioned methods would just affect the time complexity of our method and not the approximation, thus, we are not sure if it is in the scope of our work to readily have them implemented.
> > >
> > > Overall, we thank you for bringing up your concerns and helping us improve the paper. Please let us know if our response were satisfying or if you have any further concerns.

---

> ### Author Response · Authors · 2022-08-08
> **Discussion ends in less than a day**
>
> Dear reviewer,
>
> We would like to kindly send you a reminder and ask if you were satisfied with our responses below and ask if you have any further concerns or feedbacks, as the discussion window is going to end in less than a day. We would be more than happy to continue the discussion and address any further concerns that you might have.
>
> Thanks,
> Authors

---

### Official Review · Reviewer_xjB7 · 2022-07-11

**Rating:** 5
**Confidence:** 4
**Soundness:** 3 good
**Presentation:** 3 good
**Contribution:** 3 good

**Summary:**

To reduce the computational cost during active learning process, this paper proposed an interesting active learning algorithm that utilizes look-ahead strategies by large margins and approximate neural network using neural tangent kernel. So that the look-ahead strategies that can be easily adopted in classical ml tasks like EER, EMOC, could also be implemented in deep learning tasks. The authors proved that the outputs of very wide NNs can be approximated using a linearized network wit neural tangent kernels. Compared with naive computation of look-ahead strategies, this approximation is of 100 times faster and has better performance.


**Questions:**

I have several questions about this paper:

1. Figure 2 with $\Delta acc$ is really hard to read, can author just show accuracy-cycle(labeling cost) curves?

2. For implementation of LL4AL, it is very strange that LL4AL perform so bad on SVHN, CIFAR10 and CIFAR100 datasets, although the author mentioned that this is because the unstably training during AL training processes, but according to [3], the LL4AL (named LPL in [3]) perform well on SVHN, CIFAR10 and CIFAR100 datasets with ResNet18 as basic learner. I wonder did the author correctly re-implemented the LL4AL with proper way, i.e., the original version <https://github.com/seominseok0429/Learning-Loss-for-Active-Learning-Pytorch> or the re-implemented version <https://github.com/Mephisto405/Learning-Loss-for-Active-Learning>.

**I will give score 4 first and I hope the author could answer the question 2 properly. I will increase my score if I can be persuaded.**

[3] Zhan X, Wang Q, Huang K, et al. A comparative survey of deep active learning[J]. arXiv preprint arXiv:2203.13450, 2022.



**Limitations:**

The author addressed the limitations but I didn't find the discussions about potential negative societal impact of their work.

**Strengths And Weaknesses:**

Strengths: this paper is well-written with abundant proofs/experiments to theoretically/empirically demonstrate the effectiveness of their proposed models of look-ahead strategies with large margin.

Weaknesses: using NTK for approximation itself is not new in active learning, [1] use NTK approximation that construct a suitable neural embedding that determines the feature space in active learning with model selection. [2] use active learning to speed-up up the convergence for SSL deep learning algorithms, is also inspired by NTK, when the eigenvalues of the NTK are large, the convergence rate is faster.

[1] Wang Z, Awasthi P, Dann C, et al. Neural active learning with performance guarantees[J]. Advances in Neural Information Processing Systems, 2021, 34: 7510-7521.

[2] Kong S T, Jeon S, Lee J, et al. Relieving the Plateau: Active Semi-Supervised Learning for a Better Landscape[J]. arXiv preprint arXiv:2104.03525, 2021.

---

> ### Author Response · Authors · 2022-08-02
> **Response to Reviewer xjB7 (1/2)**
>
> Thank you for your comments, which will improve our paper. We respond to various specific questions below; we’ll integrate these responses into the paper in revision. If anything remains unclear or you think it needs further discussion, please do continue the conversation!
>
> > 1. Using NTK for approximation itself is not new in active learning, [1] use NTK approximation that construct a suitable neural embedding ... [2] use active learning to speed-up up the convergence for SSL deep learning algorithms...
>
> Thank you for informing us about these two papers (which we will call [A] and [B] to avoid clashing with the paper’s citations), of which we were not previously aware. Although we agree that both these approaches involve some form of computing the NTK, we would argue that both these works are totally different from ours in terms of both how they use the NTK and (especially for [A]) the problem they are trying to solve. In what follows, we go through both these works and compare them to our work where reasonable comparisons can be made.
>
> In general, both of these works consider the case of “myopic” active learning, where the the querying strategy for labeling the points is based on the current representations: to be specific, [A] falls in the “uncertainty-based” category of [3]’s Figure 1, and [B] falls in the “Representative” category. Our work, to the best of our knowledge, is the first work to enable look-ahead active learning for deep neural networks, where the querying strategy is based on how the model’s predictions change retraining with a candidate point added (not in any category of [3]’s Figure 1). Although this acquisition would result in perfect active learning, because the SGD re-training process is notoriously expensive, this hasn’t been explored in practice. By using recent theory of NTK’s approximation of the training dynamics of neural networks, we were able to approximate this retraining process and achieve state-of-the-art results for pool-based active learning.
>
>
> Paper [A] addressed the problem of streaming active learning in the non-parametric model regime, where the algorithm receives an unlabeled point and has to decide if it should be labeled or not. This is different from our task at our hand, which is pool-based active learning for a specific class of parametric models (deep neural networks). To the best of our understanding, their approach uses the NTK of a depth-n width-m MLP with ReLu activation as a general non-parametric model; our work, as typical in active learning, uses a “non-rectangular” custom architecture, say ResNet18. [A] uses the complexity of the unlabeled point at step $T$ ($S_{T,n}(h) = \\sqrt{h^\\top H^{-1} h}$ where $H$ is the mentioned NTK) to decide if the point should be labeled or not. Their Algorithm 1 is fixed to the NTK computed at initialization for a random depth-n width-m MLP, and so does not conduct any representation learning of its own – unlike our method, which uses the representation learned by the model we’re training. Thus, although if we use a “rectangular” MLP this could be a good approximation at initialization, over the course of training it will quickly become very different from the model we’re actually training (like our experiments with an infinite NTK, although even worse because the architecture is different). This would enforce the provided rates to be vacuous, as the NTK’s condition number would grow linearly with number of training points.  Although the authors propose to update the weights in Appendix A.3 using one gradient step (not ideal), this makes the approach practically infeasible, since they recompute the NTK matrix after each step.
>
> Paper [B] aims at solving a specific case of pool-based active learning, where the unlabeled pool is separated into multiple groups. Their idea is based on the connection between the eigen-spectrum of the model’s NTK and trainability of the model, which has been recently explored in the literature.  Although this approach sounds theoretically promising, in practice, for every new data point, they need to find the minimum eigenvalue of the empirical NTK, which requires $\\mathcal O(U L^3)$ time where U and L denote the number examples in the unlabeled and labeled pools, respectively. This is 1) far more expensive than our acquisition strategy, and 2) not a look-ahead criterion, thus not bearing our expectation of “perfect” active learning when the approximation is exact. This is also much slower than the $O(L^3)$ (with naive matrix inversion method) of our method, which we think is why they did experiments with only few hundred points (their Table 1 - 3); in that low-budget regime, they need SSL techniques to have reasonable performance. Moreover, they have to form subspaces of the unlabeled data (i.e. groups) to reduce the time complexity, which as they have discussed, has its own cons and deviates from the standard pool-based active learning setting.

---

> ### Author Response · Authors · 2022-08-02
> **Response to Reviewer xjB7 (2/2)**
>
> > 3. For implementation of LL4AL, it is very strange that LL4AL perform so bad on SVHN, CIFAR10 and CIFAR100 datasets, although the author mentioned that this is because the unstably training during AL training processes, but according to [3], the LL4AL (named LPL in [3]) perform well on SVHN, CIFAR10 and CIFAR100 datasets with ResNet18 as basic learner. I wonder did the author correctly re-implemented the LL4AL with proper way, i.e., the original version https://github.com/seominseok0429/Learning-Loss-for-Active-Learning-Pytorch or the re-implemented version https://github.com/Mephisto405/Learning-Loss-for-Active-Learning.
>
> We understand the concern about the poor performance of LL4AL. We were also quite surprised by these results, and put in a significant amount of effort trying to identify the root of the difference. Although the reported performance of LL4AL is significantly better than reported random acquisition baselines, we have observed that this mostly happens when the learning rates used in the process of training the models are fairly low, with the same learning rates are used for the random baseline.
>
> As reported in many previous papers (e.g. https://arxiv.org/abs/1708.07120), high learning rates tend to enhance the performance of overparameterized neural networks on the test-set. For fair comparison between all acquisition methods while maximizing the performance of the base model, we use the OneCycle learning rate scheduler, which has been shown to achieve far better evaluation-time performance than other schedulers especially on residual networks (same paper). For each network-dataset pair, we find the learning rate that maximizes the performance of the random acquisition strategy, then use the same learning rate for all other acquisitions to be fair across all approaches. Under this strategy, we observe that random acquisition someti performs better than what is reported in recent active learning papers. Most other acquisition functions we tried also improve in this regime, but LL4AL’s performance decreases substantially. We suspect that when the learning rates are not tuned to maximize the baseline model’s performance, LL4AL’s loss module helps fill in the gap and make up for the non-optimal learning rate, but when the learning rates are fully-tuned, the alteration in the loss function is detrimental to LL4AL’s performance. Deeper investigation of this phenomenon, though, is out of the scope of the present work.
>
> Nevertheless, to provide more evidence, we compare the random acquisition baseline to LL4AL using the implementation in https://github.com/Mephisto405/Learning-Loss-for-Active-Learning with higher learning rates. The results (averaged over three runs) are added to **Appendix E** in the revised version; we will expand the discussion of this phenomenon in the appendix in revision.
>
>
> > 4. The author addressed the limitations but I didn't find the discussions about potential negative societal impact of their work.
>
> We are not aware of any particular potential negative impact of our work compared to other pool-based active learning approaches, a well-established sub-field of machine learning research that doesn’t seem particularly harmful in itself, though we will add a mention of being careful about potential misuse of active learning relative to human decision-making. We would be happy to hear any potential negative impact of our work in particular, or any work to cite on problems with active learning in general, that we might have missed.

---

> ### Author Response · Authors · 2022-08-04
> **On LL4AL's performance**
>
> In order to further clarify and investigate LL4AL's performance and stability under different learning rates, we have anonymously shared our further experiments based on the LL4AL re-implementation code as suggested by you in this repository: https://github.com/anonymous-ml-enjoyer/ll4al-reproduce-lr
>
> Here, we further clarify our observations based on the plots generated by the diverse experiments conducted using the code above (reported numbers are average of 3 different seeds):
> * **CIFAR10, ResNet18**:
>   * We observe that although when using the learning rate of 0.1 LL4AL's performance is ~4% better than random (91.3% vs 87.3%), when the learning rate is tuned in favour of random (0.3 LR instead of 0.1) and use that for LL4AL too, the difference shrinks down to 1.2% (90.0% vs 87.8%). Moreover, using larger learning rates like 1.0 would result in LL4AL to diverge while random still gets descent results.
> * **SVHN, ResNet18**:
>   * Again, the difference is 3.3% (93.8% vs 90.5%) when LR is tuned in favour of LL4AL, but drops to 1.1% (91.9% vs 90.8%) when using a LR of 1.0. This time however, when LR is fully tuned in favour of random (2.0), LL4AL diverges.
> * **CIFAR10, 1-block WideResNet**:
>   * The 2.6% difference when using 0.1 LR (82.2% vs 79.6%)  drops to -1.8% (78.8% vs 80.6%) when LR is tuned in favour of random (0.3). We remind that as we're using one block layer here, the LL4AL's performance drop is much more noticeable to the point that it performs worse than random when LR is tuned in favour of random!
> * **SVHN, 1-block WideResNet**:
>   * The 2% difference when using 0.1 LR (95% vs 93%) drops to -1.4% (92.4% vs 93.8%) when LR is tuned in favour of random (0.5).
>
> All these experiments are also showcased in **Appendix E** of the revised version.
>
> We hope that this batch of experiments, which is directly based on the reviewer's suggestion, solves the confusion. Meanwhile, we would be happy to conduct any further experiment or hear related feedbacks that could help further clarify the case and improve the manuscript and the community's understanding of this method.

---

> ### Author Response · Authors · 2022-08-07
> **Discussion ends in less than 3 days**
>
> Dear reviewer,
>
> Since you indicated that
>
> > I  will give score 4 first and I hope the author could answer the question 2 properly. I will increase my score if I can be persuaded.
>
> we wanted to inquire whether you were satisfied with our responses below, both regarding the related work on using NTK in active learning and LL4AL's performance. If not, please let us know soon so that we could provide more details or address your other questions/concerns before the discussion period ends.
>
> Moreover, we would like to let you know that according to your suggestion on your first point, we have added the accuracy-cycle (labeling cost) curves to the **Appendix J** of the current revision (which is uploaded and is accessible in the supplementary material).
>
> Thank you for your time and attention!

---

> > ### Comment · Reviewer_xjB7 · 2022-08-07
> > **response**
> >
> > Dear authors,
> >
> > I would like to thank the authors for their clarifications and making revisions of the appendix, I have checked it. The findings of LL4AL's performance and the learning rate is indeed interesting, It's good. But after read the appendix, I have further questions, did you conduct t-test of your proposed approach with AL baselines like Entropy? Since at least in Figure 8b, 8c, 8e, 8f and 8g, the accuracy-budget curves of NTK and Entropy are quite similar.

---

> > > ### Author Response · Authors · 2022-08-07
> > > **Thanks for the quick response.**
> > >
> > > Thank you for the response and the question! Your observation is right. For those pairs of models and datasets, the proposed NTK method is not significantly better than strong baselines. Likewise, we did not intend to claim that our method outperforms the state-of-the-art performance in pool-based active learning, and we mentioned that it either slightly outperforms or achieves the same performance in comparison to the state-of-the-art methods. We believe that the significance of our work mostly comes from the fact that this is the first work that enables look-ahead strategies in deep active learning (through reducing the computational needs of retraining by using the proposed local NTK approximation), and readily performs **comparable** (sometimes even better) to the state-of-the-art active learning methods.
> > > Due to high computational cost of these strategies, previous look-ahead approaches rely on special model classes such as Na\”ive Bayes or Gaussian process which do not perform as good as neural networks.
> > >
> > > The table below shows paired t-test on look-ahead acquisitions and strongest baselines: Entropy and Margin, on MNIST with a 1-layer WideResNet as done in Appendix I. The previous look-ahead approaches: NN 1-step, NN Inf NTK and GP, are all significantly worse than NN baselines: Entropy, Margin, BADGE and LL4AL, whereas there is no significant difference between the proposed NTK approximation and the baselines (Entropy: $0.7960 > 0.05$, Margin: $ 0.4539 > 0.05$, BADGE: $0.5805 > 0.05$, and LL4AL: $0.0590 > 0.05$).
> > > Please note that this is not just the case between the NTK and the baselines. There is no statistically significant difference when comparing these baselines with themselves (i.e. BADGE and LL4AL) either.
> > >
> > >
> > >
> > > > methods        |   Entropy   |   Margin   |  BADGE  |  LL4AL  | NN NTK   |   NN 1-step   |   NN Inf NTK   |   GP    |
> > > > --------------|--------------|--------------|--------------|--------------|--------------|--------------|--------------|--------------|
> > > > Entropy         |     1.0000    |    0.4896     |  0.5698  |  0.1434   |   0.7960            |    0.0287   |   0.0226   |  0.0042   |
> > > > Margin          |     0.4896    |    1.0000        |  0.8534  |  0.2256  |    0.4539           |    0.0284   |   0.0219    |  0.0030  |
> > > > BADGE          |  0.5698  |  0.8534  |   1.0000        |    0.1748          |    0.5805   |   0.0239    |  0.0187  |  0.0027  |
> > > > LL4AL          |     0.1434  |  0.2256  |  0.1748  |  1.0000        |    0.0590           |    0.0911   |   0.0617    |  0.0056  |
> > > > NN NTK        |    0.7960    |     0.4539       |   0.5805  |  0.0590  |  1.0000           |   0.0109     |   0.0093  |   0.0014    |
> > > > NN 1-step     |    0.0287   |      0.0284       |  0.0239  |  0.0911  |    0.0109           |   1.0000     |   0.6863  |   0.0422   |
> > > > NN Inf NTK |   0.0226    |     0.0219        |   0.0187  |  0.0617  |  0.0093         |   0.6863    |    1.0000  |    0.0742   |
> > > > GP                  |    0.0042   |     0.0030      |  0.0027  |  0.0056  |    0.0014         |   0.0422     |    0.0742  |    1.0000   |
> > >
> > > As suggested, we will add the paired t-test results on different combinations in the appendix. Please let us know if our response answers your question or if you have any other questions/concerns.

---

> > > > ### Comment · Reviewer_xjB7 · 2022-08-08
> > > > **response**
> > > >
> > > > But if your work just aims to apply look-ahead strategies in DL tasks and better than previous look-ahead stratgies, the contribution seems somewhat not enough. Since the work belongs to active learning field. Can authors provide a deeper description of the contributions of this work?

---

> > > > > ### Author Response · Authors · 2022-08-08
> > > > > **Thanks for the quick response.**
> > > > >
> > > > > Thank you for the response and mentioning your concern. As requested, here is the summary of our contributions.
> > > > >
> > > > > Contributions:
> > > > >
> > > > > 1. We have proved that cold-start and warm-start in deep active learning are asymptotically equivalent (Theorem 3.1 and Appendix A).
> > > > > 2. Based on this, we have proposed an algorithm that can approximate SGD retraining of deep neural network models using a local linearization and kernel augmentation, which enables using any look-ahead acquisition functions in deep active learning (Section 3). We have also shown that the error of approximation is bounded under reasonable assumptions (Appendix K), which is empirically supported (Appendix H).
> > > > > 3. We have demonstrated that the proposed method significantly outperforms previous look-ahead acquisition strategies that were applied on models other than neural networks in terms of both performance (Figure 1b and Appendix I) and computational time (Figure 1), and also gives comparable or even better performance compared to the-state-of-the-art methods (Figure 2 and Appendix D, J).
> > > > > 4. The proposed method also enables sequential query strategy which has not been available for previous active learning methods (Figure 3a).
> > > > > 5. The proposed method can potentially be a framework for further developments on look-ahead strategies in deep active learning.
> > > > >
> > > > > Regarding the comparison with the state-of-the-art methods, we want to emphasize that the proposed NTK method is mostly better than the state-of-the-art methods as shown in Figure 2 and 4. Moreover, in comparison to recent proposed methods, it is not suffer from being unstable against learning rate (like LL4AL) and achieves mostly better results than other recently proposed ones (like BADGE) without having any hyperparameters. We just did not claim that it is so much better that it is **statistically significantly** better than them with the mild number of evaluations that can be reasonably run.
> > > > >
> > > > > We also want to emphasize that look-ahead acquisition strategies are the strategies that truly reflect how a model would actually behave on a new data point, and in that sense, it is much closer to the ultimate goal of active learning than uncertainty or representation-based strategies. Despite their potential to be another main branch of active learning along with uncertainty and representation-based strategies, look-ahead acquisition strategies have not been well studied (2 papers in the past 5 years to the best of our knowledge) due to extremely high computational needs. It is not a coincidence that they have not been studied in recent years: neural networks dominate the field but look-ahead acquisition strategies were not applicable to neural networks.
> > > > >
> > > > > We believe our work can boost up further studies on look-ahead strategies, which will expand possible research directions in deep active learning by the community.

---

> > > > > > ### Comment · Reviewer_nYBt · 2022-08-08
> > > > > > **This comment caused me to further increase my score**
> > > > > >
> > > > > > Dear authors,
> > > > > >
> > > > > > Thanks. Especially:
> > > > > >
> > > > > > >  We just did not claim that it is so much better that it is statistically significantly better than them with the mild number of evaluations that can be reasonably run. Please note that according to our experiments, none of the existing methods meet this bar compared to one another in most settings.
> > > > > >
> > > > > > I think it would be very welcome if you can make this small addition (or add some emphasis on this?) in your paper somewhere - I definitely missed it.. I think it is very important to mention this; it may not always be clear from the deep active learning literature. In fact, I would recommend you to provide a bit more detail regarding the statistical test you ran and those results (somewhere in the appendix somewhere?). This is also an important fact for the community to generally realize. Active learning gains, I unfortunately think, are somewhat overstated sometimes.
> > > > > >
> > > > > > > We also want to emphasize that look-ahead acquisition strategies are the strategies that truly reflect how a model would actually behave on a new data point, and in that sense, it is much closer to the ultimate goal of active learning than uncertainty or representation-based strategies. Despite their potential to be another main branch of active learning along with uncertainty and representation-based strategies, look-ahead acquisition strategies have not been well studied (2 papers in the past 5 years to the best of our knowledge) due to extremely high computational needs. It is not a coincidence that they have not been studied in recent years: neural networks dominate the field but look-ahead acquisition strategies were not applicable to neural networks.
> > > > > >
> > > > > > I think this is also a great argument in favour of your work. I also understand you will publish your code to go with it? I think that will indeed be very valuable. Thanks!

---

> > > > > > > ### Author Response · Authors · 2022-08-09
> > > > > > > **Thank you**
> > > > > > >
> > > > > > > Dear reviewer,
> > > > > > >
> > > > > > > We appreciate the valuable and encouraging feedback.
> > > > > > >
> > > > > > > > I think it would be very welcome if you can make this small addition (or add some emphasis on this?) in your paper somewhere - I definitely missed it.. I think it is very important to mention this; it may not always be clear from the deep active learning literature. In fact, I would recommend you to provide a bit more detail regarding the statistical test you ran and those results (somewhere in the appendix somewhere?). This is also an important fact for the community to generally realize. Active learning gains, I unfortunately think, are somewhat overstated sometimes.
> > > > > > >
> > > > > > > Our observations based on the results of our experiments on the other combinations of models and datasets also confirm that although the different deep active learning methods that we have tested result in different performances, their difference has mostly not been statistically significant. We will add the suggested Friedman’s test and post-hoc paired t-tests on other combinations of datasets and model architectures in the appendix along with detailed explanations as we did on the comment above, and will refer to it in the main body.
> > > > > > >
> > > > > > > > I think this is also a great argument in favour of your work. I also understand you will publish your code to go with it? I think that will indeed be very valuable. Thanks!
> > > > > > >
> > > > > > > We will definitely make the code public with clear instructions on running the code and the detailed configuration files that we used to run the experiments. Meanwhile, if any reviewer wants to check our implementation, we recommend checking the anonymous code we provided in the supplementary material.
> > > > > > >
> > > > > > > Thanks,
> > > > > > > Authors

---

> > > > > > ### Comment · Reviewer_xjB7 · 2022-08-09
> > > > > > **new comment**
> > > > > >
> > > > > > The summarization of the contribution is good, it helps to understand the paper, the author should revise the main paper according to this statement.
> > > > > >
> > > > > > However, it still remain concerns. Look at this:
> > > > > >
> > > > > > > We just did not claim that it is so much better that it is statistically significantly better than them with the mild number of evaluations that can be reasonably run. Please note that according to our experiments, none of the existing methods meet this bar compared to one another in most settings.
> > > > > >
> > > > > > It is not appropriate, since SOTA methods already exists like WAAL [w1]. WAAL is a strong threorical supported AL method and works far better than normal AL methods on multiple various tasks, can see more experimental results on [w2]. So it is the author failed to compared their work with SOTA methods, not *none of the existing methods meet this bar compared to one another in most settings*. The authors should revise their statement, give a more convincing statement to explain "although the proposed method perform on par with baselines, but it still has its advantages" or add new experiments.
> > > > > >
> > > > > > The keypoint is, when a new researcher needs to select an AL method for some new tasks, why he/she choose NTK, not CONF? not Margin? not Entropy? Can author revise their previous statement along this point?
> > > > > >
> > > > > > [w1]  Shui C, Zhou F, Gagné C, et al. Deep active learning: Unified and principled method for query and training[C]//International Conference on Artificial Intelligence and Statistics. PMLR, 2020: 1308-1318.
> > > > > >
> > > > > > [w2] Zhan X, Wang Q, Huang K, et al. A comparative survey of deep active learning[J]. arXiv preprint arXiv:2203.13450, 2022.

---

> > > > > > > ### Comment · Reviewer_nYBt · 2022-08-09
> > > > > > > **It is not always about improving the SOTA**
> > > > > > >
> > > > > > > This work has potential to develop into many new active learning strategies based on look ahead. It is making look-ahead strategies viable for neural networks, generally. Therefore, this is likely a high impact work, on which the community can further iterate. In that light, I think it is fine that it does not beat the state of the art.
> > > > > > >
> > > > > > > Anyway; I am very suspicious about any claims of state-of-the-art results in deep active learning... Especially since the field is so new, there are no agreed upon standards for the experimental design. So any claims of the state-of-the-art should be taken with a grain of salt, in my opinion.

---

> > > > > > > > ### Comment · Reviewer_xjB7 · 2022-08-09
> > > > > > > > **further response**
> > > > > > > >
> > > > > > > > I agree with your second point, there is no agreed upon standard for the experimental design. But your previous statement is inappropriate, when new readers read your paper, they won't regard it as a good contribution/advantage. Your first point is good, can you write a revised version of contribution based on the first point?

---

> > > > > > > > > ### Author Response · Authors · 2022-08-09
> > > > > > > > > **Thanks for the responses to both reviewers.**
> > > > > > > > >
> > > > > > > > > Dear reviewer xjB7,
> > > > > > > > >
> > > > > > > > > Thanks for your response and the feedback on our summarization. We also thank reviewer nYBt for engaging in the conversation and their valuable point that they made. We would like to further clarify our responses while addressing your concerns below:
> > > > > > > > >
> > > > > > > > > > It is not appropriate, since SOTA methods already exists like WAAL [w1]. WAAL is a strong threorical supported AL method and works far better than normal AL methods on multiple various tasks, can see more experimental results on [w2]. So it is the author failed to compared their work with SOTA methods, not none of the existing methods meet this bar compared to one another in most settings. The authors should revise their statement, give a more convincing statement to explain "although the proposed method perform on par with baselines, but it still has its advantages" or add new experiments. The keypoint is, when a new researcher needs to select an AL method for some new tasks, why he/she choose NTK, not CONF? not Margin? not Entropy? Can author revise their previous statement along this point?
> > > > > > > > >
> > > > > > > > > Although we do agree that WAAL's performance is much better than other acquisition strategies on the mentioned tasks, we do not agree that it's fair to compare WAAL to other methods like LL4AL, BADGE, Entropy, our proposed method or similar approaches that do not use the unlabeled data in the process of training the model. As also seen on [w2], WAAL is categorized under a different branch of acquisitions. As such, we agree that WAAL is a strong method with solid experimental results, but it's aiming at solving a (slightly) different problem than the task at our hand, and should be compared with other semi-supervised active learning methods that use the unlabeled pool in the process of training the model (please refer to question 3 of reviewer gpbw).
> > > > > > > > >
> > > > > > > > > Moreover, we would like to clarify that we did not have a general claim regarding the experiments done in the literature not showing any significant advantage of any method over any other one, rather, we claimed that **in the set of experiments that we have run** (whose results are available in the according plots in the main body and appendix -- for which we will also add the corresponding hypothesis test results), we found that no method is statistically significantly superior to other ones among the recent SOTA methods, although some achieve better accuracies (i.e. entropy and NTK tend to have solid performances). We tried to cover all the methods that solved similar task to ours in the experiments, but we would have been happy to further run experiments involving any other fairly comparable method that you might suggest, if time allowed.
> > > > > > > > >
> > > > > > > > > Lastly, we would like to thank you for your valuable suggestion about adding a statement that clarifies:
> > > > > > > > >
> > > > > > > > > > "although the proposed method perform on par with baselines, but it still has its advantages"
> > > > > > > > >
> > > > > > > > > We will incorporate this and revise the main body of the paper accordingly to reflect this point with greater details and statements regarding the significance of enabling the look-ahead approach in deep active learning, as reviewer nYBt suggested.
> > > > > > > > >
> > > > > > > > > > This work has potential to develop into many new active learning strategies based on look ahead. It is making look-ahead strategies viable for neural networks, generally. Therefore, this is likely a high impact work, on which the community can further iterate. In that light, I think it is fine that it does not beat the state of the art.
> > > > > > > > >
> > > > > > > > > Finally, we thank both reviewers for their positive discussion that will helps us improve the paper.

---

> > > > > > > > > > ### Comment · Reviewer_xjB7 · 2022-08-09
> > > > > > > > > > **raise score**
> > > > > > > > > >
> > > > > > > > > > I would raise the score to 5. Please further refine your papers.

---

> > > > > > > > > > > ### Author Response · Authors · 2022-08-09
> > > > > > > > > > > **Thanks for the feedback**
> > > > > > > > > > >
> > > > > > > > > > > Dear reviewer xjB7,
> > > > > > > > > > >
> > > > > > > > > > > Thank you for recognizing our contribution. We have revised our previous comment as you suggested: 1) we added another contribution that our work can potentially be a framework for further developments on look-ahead strategies in deep active learning, 2) we took out “Please note that according to our experiments, none of the existing methods meet this bar compared to one another in most settings.” as it can be misinterpreted as you pointed out. We will reflect them in the main paper.
> > > > > > > > > > >
> > > > > > > > > > > Thank you for your constructive feedback.

---

### Official Review · Reviewer_nYBt · 2022-07-13

**Rating:** 7
**Confidence:** 4
**Soundness:** 3 good
**Presentation:** 2 fair
**Contribution:** 3 good

**Summary:**

The work introduces a new active learning technique, that aims to use the model uncertainty over the pool of unlabeled data, after the model is retrained on a new additional point on the pool (so not a myopic strategy - but they call this a "look ahead strategy"). To make the strategy computationally feasible, approximations based on the neural tangent kernel are used.

**Questions:**

7) In Figure 1b; how can it be that GP (Random) is performing better than GP? This must be wrong... It seems like the baseline was modified - were these modifications sensible? Would it be possible to run the experiments with the original GP for the classification tasks? I actually wonder how that would compare... It is

8) In the figures, there is a shaded region indicating variability, but it is not explained how this is calculated or what this is? How many runs were used? Is this the standard error or standard deviation? Are the differences between the methods significant?  Since it is claimed that "existing look-ahead strategies by large margins," I would expect to see some significant p-values from a paired t-test comparing methods pairwise (as a posthoc test after a Friedman test).

9) The query scheme, I don't understand... First of all, the figures mention "Cycles" - why not just indicate the amount of training data available? It makes it unclear. Second, the cycles seem only defined up to 4 or 5. What happens after that? This is not specified in the "query scheme" (line 280). On the y-axis sometimes is given "\Delta Accuracy"; what is that and how is that defined?

10) The analysis and derivation of equations 6-8 seems to share many similarities with the derrivation of the EMOC strategy itself. What are the similarities? Was your work inspired by their work? Please indicate the similarities and give proper credit.



**Limitations:**

Please see above.

**Strengths And Weaknesses:**

1) The work introduces a technique based on an idealized criteria, which is then approximated to make it computationally feasible. However, the approximations are changed in such a way, that the idealized criteria is not being approximated anymore. This weakens the story, as we cannot explain what the active learning strategy is doing, and why it works. The performance boost seems now to come from somewhere else, but the reasoning is unclear why this is the case. Various heuristics are implemented but there is not a careful ablation study to verify where the performance gains are coming from.

2) It is claimed the proposed look ahead strategy MLMOC improves matters. However, a detailed comparison with other already published lookahead strategies is missing, such as the original EMOC strategy. It is claimed that EMOC didn't work well, but no experiments are given to back up this claim. Similarly, several ad-hoc changes are made (line 208-210), which are claimed to "empirically not hurt accuracy" but this is not shown in the paper. Why are these results withheld?

3) Some other claims that are made; which are not backed up by evidence:
3a) "We also suspect that, when ... is already reasonably accurate, we can “trust” the most likely label more than we can trust accurate estimation of probabilities for low-probability losses, especially when training networks with..." (line 210).  Can we not check this?
3b) "These classes of models, however, tend to not work as well as modern neural networks, meaning these approaches are outperformed by simpler acquisition functions on stronger models." Can we check this with a Gaussian Process classification baseline with a GP acquisition strategy? E.g. compare with [5] or [6]? (line 38)
3c) "Sequentially adding true labels of new data gives performance substantially better than more common batch setups" [line 58] Source? I think actually, the batch strategies can sometimes outperform sequential strategies, especially for myopic strategies that do not take diversity into account, while a batch adaptation usually does check this.

4) Since the approximation allows (I think) various variants of the EMOC criteria, I was expecting to also see a comparative study on this aspect, for example GP-impact, EMOC, and some of the other baselines in from [5,6].

5) The experimental setup is not carefully described and not using the standards of the active learning field (e.g. check Settles). Check q9 for more detail.

6) Strength: the approximations introduced to make look-ahead strategies feasible using the Neural Tangent Kernel, which I think is an important contribution, since it can be used to design various new active learning strategies.

---

> ### Author Response · Authors · 2022-08-02
> **Response to the Reviewer nYBt  (1/4)**
>
> Thank you for your feedback, which will improve our paper; we’ll incorporate the various new results and clarifications given here in revision. If anything remains unclear or you think it needs further discussion, please do continue the conversation! Respectfully, we abbreviated the questions to have more space for focusing on the response.
>
> > 1. The work introduces a technique based on an idealized criteria, which is then approximated to make it computationally feasible. However, … there is not a careful ablation study to verify where the performance gains are coming from.
>
> We make two such approximations/heuristics in this work. The first is to approximate the retraining process with the NTK; see our separate comment on the quality of this approximation. The other heuristic is the MLMOC criterion, discussed next. We believe these choices have good justifications and experimental evaluation; please let us know if you think there are other significant approximations or gaps we should be exploring further.
>
>
> > 2. It is claimed the proposed look ahead strategy MLMOC improves matters. However, a detailed comparison with other already published lookahead strategies is missing, … several ad-hoc changes are made (line 208-210), which are claimed to "empirically not hurt accuracy" but this is not shown in the paper. Why are these results withheld?
>
> Thank you for the valuable suggestions. To be clear, we _did not_ intend to claim that MLMOC gives better active learning performance than EMOC: we use MLMOC over EMOC because it saves computational cost, proportional to the number of classes, without losing out on active learning quality. We indeed should have included these results in the paper; we have added this to **Appendix G** in the current revision which compares MLMOC to EMOC, EER, and other acquisition functions.

---

> > ### Comment · Reviewer_nYBt · 2022-08-05
> > **Thanks**
> >
> > I would like to thank the authors for their clarifications, I am mostly satisfied and actually have no additional comments.

---

> ### Author Response · Authors · 2022-08-02
> **Response to the Reviewer nYBt (2/4)**
>
> > 3. Some other claims that are made; which are not backed up by evidence:
>
> > 3a) "We also suspect that ..., we can “trust” the most likely label more than we can trust accurate estimation of probabilities for low-probability losses,..." (line 210). Can we not check this?
>
> As we mentioned, this statement was speculation; given that the true conditional distribution of a label given an input dataset is not known, we are not sure of a way to evaluate this statement (we’d be happy to hear suggestions!), but it is not important to our argument and are happy to remove it.
>
> > 3b) "These classes of models, however, …, are outperformed by simpler acquisition functions on stronger models." Can we check this with a Gaussian Process classification baseline with a GP acquisition strategy? E.g. compare with [5] or [6]? (line 38)
>
> The methods in [5, 6] are based on handcrafted features e.g., RBF kernels on bag-of-visual-words of SIFT featurizations. We did in fact compare to a GP model in Figure 1b, showing that a neural network with random acquisition (NN Random) outperforms a GP using the infinite NTK with either random acquisition (GP Random) or EMOC (GP). We did, as requested, also compare to these SIFT-RBF kernels, showing that indeed they are substantially worse than a neural network with random acquisitions:
>
>
> > cycle (number of labeled data) |  0 (100)  |  1 (120)  |  2 (140)  |  3 (160)  |  4 (180)  |  5 (200)  |  6 (220)  |  7 (240)  |  8 (260)  |  9 (280)  |
> > ---------------------------------|--------|--------|--------|--------|--------|--------|--------|--------|--------|--------|
> > GP RBF (EMOC)         | 0.5019 | 0.5158 | 0.5283 | 0.5452 | 0.5594 | 0.5677 | 0.5787 | 0.5602 | 0.6107 | 0.6139 |
> > GP  inf NTK (EMOC) | 0.8193 | 0.8266 | 0.8315 | 0.8399 | 0.8421 | 0.8505 | 0.8507 | 0.8567 | 0.8760 | 0.8760 |
> > NN Random                 | 0.8770 | 0.9306 | 0.9513 | 0.9586 | 0.9589 | 0.9576 | 0.9652 | 0.9623 | 0.9644 | 0.9659 |
>
> More importantly, we think that the baselines in [6] including GP-impact are not look-ahead strategies, and so are not particularly relevant to the statement here. Only EER [5] and EMOC [6] are look-ahead strategies, thus, they are irrelevant to the sentences the reviewer referred to (please refer to the answer for details about GP-impact).
>
> EER using Naive Bayes [5] also focuses on binary classification on text datasets; modifying it to be suitable for multi-class classification on image datasets is substantially out of scope for this work, especially given that the last ten years of progress in machine learning have (we think) fairly substantially demonstrated that modern neural networks are far more powerful than methods like Naive Bayes on image datasets. We do think that comparing MLMOC to EMOC and EER based on the same neural network structure is far more meaningful, as we did in question 2.
>
>
> > 3c) "Sequentially adding true labels of new data gives performance substantially better than more common batch setups" [line 58] Source?
>
> We think there is some confusion here about what we’re referring to with “sequentially adding true labels.” It is indeed the case that greedily building a batch to query simultaneously can be significantly worse than batch-based methods (when the acquisition function is not submodular). However, here we’re talking about something else: choosing a point to query, immediately observing its true label, then choosing the next point to query, i.e. using a batch size of 1 rather than a large batch. This provides strictly more information than batch-based regimes, regardless of how batches are constructed.
>
> In traditional active learning setups, this is of course not a fair comparison to batch regimes. Our point here is that when the requirement for batch querying is not because of the inherent data regime (e.g. we want to run several long lab experiments in parallel) but rather due to the delay in retraining the underlying neural network, our method allows for a regime where we do observe the true label point-by-point without retraining after each step, unlike most related methods. We will try to clarify this further in the text.

---

> ### Author Response · Authors · 2022-08-02
> **Response to the Reviewer nYBt (3/4)**
>
> > 4. Since the approximation allows (I think) various variants of the EMOC criteria, I was expecting to also see a comparative study on this aspect, for example GP-impact, EMOC, and some of the other baselines in from [5,6].
>
> We are unsure what you mean by “various variants of the EMOC criteria,” but as far as we understand, among the methods used in [5, 6], only EMOC and EER are look-ahead acquisition strategies. For instance, GP-impact only considers how a GP prediction on a new data point would change when it is added to the labeled set, which is an ``expected model change’’ type of acquisition, not a look-ahead acquisition that we focus on. We have compared the look-ahead acquisitions EMOC, MLMOC, and EER in question 2.
>
>
> > 5. The experimental setup is not carefully described and not using the standards of the active learning field (e.g. check Settles). Check q9 for more detail.
>
> If there are any specific details (other than those we address now) that you think are missing or not described in the standards of the field, please let us know so we can address them.
>
> > 9. First of all, the figures mention "Cycles" - why not just indicate the amount of training data available? It makes it unclear.
>
> We did this to be clear about when label acquisition and retraining occur, which is also done e.g. by [39] and Joshi et al. (“Multi-Class Active Learning for Image Classification”). The number of training points per cycle is given in line 283-284, but we will change the figure labels to give numbers of training points.
>
> > Second, the cycles seem only defined up to 4 or 5. What happens after that? This is not specified in the "query scheme" (line 280).
>
> Perhaps line 283 was unclear: for MNIST, we begin with 100 random points (cycle 0) and then each cycle adds 20 more points, so that cycle 1 has 120 and cycle 9 has 280. We’ll rephrase in the revision.
>
> > On the y-axis sometimes is given "\Delta Accuracy"; what is that and how is that defined?
>
> This is the difference between the accuracy of each method versus that of random acquisition at the same point (which is why the “Random” line is always flat at 0); this was explained at line 313, describing the first plot where it is used. We did this in cases where the accuracy of all methods increases significantly over the course of learning but the difference between methods is much smaller (as typical in active learning, including BADGE [4], LL4AL [38], BatchBALD [30]). This can make it very difficult to see the differences between different methods when the y-axis is accuracy. We did use plain accuracy in cases like Figure 1 and 3a, where the differences are still easy to see. However, as suggested, we will also add plots with accuracy in the Appendix following the convention.
>
> > 7. In Figure 1b; how can it be that GP (Random) is performing better than GP? This must be wrong... It seems like the baseline was modified - were these modifications sensible? Would it be possible to run the experiments with the original GP for the classification tasks? …
>
> [14] approximates EMOC using a 1-step gradient update, included in Figure 1b as NN 1-step. We are not aware of any previous usage of EMOC computation with a full retrained model for multi-class classification: [15] use EMOC for regression tasks, and [6] use it for binary classification tasks with a slight modification of GP regression. We modify the GP for multi-class classification, but again, there is no previous work establishing that EMOC should work for general multi-class classification on image datasets, either empirically or theoretically; we empirically show that it is actually worse than random acquisition baseline with GP.

---

> ### Author Response · Authors · 2022-08-02
> **Response to the Reviewer nYBt (4/4)**
>
> > 8. In the figures, there is a shaded region indicating variability, but it is not explained how this is calculated or what this is? How many runs were used?… I would expect to see some significant p-values from a paired t-test comparing methods pairwise (as a posthoc test after a Friedman test).
>
> As we stated in Line 317 - 318, each experiment is run for six different seeds, and shading shows a 95% confidence interval for the mean, by which we mean $\pm 1.96$ times the standard error (we’ll clarify).
>
> We also ran the suggested test for the existing look-ahead strategies in Figure 1b: Ours (NN NTK), [14] (NN 1-step), [40] (NN Inf NTK), and [15] (GP).
>
> Friedman’s test (using friedman function in pingouin package)
>
>
> > W            | ddof1    | ddfo2    | F       | p-unc         |
> > -------|-------|-------|-------|-------|
> > 0.911111 | 2.33     | 4.67       | 20.5 | 0.004498 |
>
> As the p-value (p-unc) is $ < 0.05$, we did a post-hoc paired t-test.
>
> Post-hoc paired t-test (using posthoc_ttest function in scikit_posthocs package)
>
>
> > methods        |   NN NTK   |   NN 1-step   |   NN Inf NTK   |   GP           |
> > --------------|--------------|--------------|--------------|--------------|
> > NN NTK        |    1.0000     |   0.0109         |   0.0093            |   0.0014    |
> > NN 1-step     |    0.0109     |   1.0000         |   0.6863            |   0.0422   |
> > NN Inf NTK |    0.0093    |   0.6863        |    1.0000            |    0.0742   |
> > GP                  |    0.0014     |   0.0422        |    0.0742            |    1.0000   |
>
> As shown in the table above, our proposed NN NTK is significantly better than the other look-ahead methods.
>
> [9 was addressed under 5.]
>
>
>
> > 10. The analysis and derivation of equations 6-8 seems to share many similarities with the derrivation of the EMOC strategy itself. What are the similarities? Was your work inspired by their work? Please indicate the similarities and give proper credit.
>
> We would like to take this opportunity to mention that our proposed NTK approximation is a general framework that can be used for any look-ahead acquisition strategies including EER, EMOC, and MLMOC, not just for EMOC. As noted, the form of equations in GP EMOC [6, 14] and our NTK approximation in (11) look somewhat similar; this is simply due to the equivalence of infinite NTKs to kernel regression and the equivalence of kernel regression with GP posterior means. We do not claim this derivation as a particularly novel contribution of our paper, but essentially just folklore. The contribution of our paper is instead to make a close approximation for look-ahead acquisition functions for a large class of neural networks, which is known to significantly outperform GPs. Hence, the motivation of our work and GP EMOC are completely different.
>
> Overall, we thank you for your valuable suggestions, concerns and points that you made. We would be happy to further discuss any possible question or feedback that you might have on our responses to further address possible concerns and enhance the quality of the work.

---

### Author Response · Authors · 2022-08-02
**Comment on approximation quality**

We’d like to comment, since a few reviewers asked about this, on the quality of approximation of our linearization to actual retraining. This is a good question, and we will add this discussion and new experiment to the paper.

Theoretically, it has been shown that the predictions of the linearized network and the actual neural network are bounded relative to the inverse of square root of the width of the network, as stated in Equation (9) of our paper.
In practice, several works have shown that this local linearization’s performance is very close to the vanilla neural network at hand – and, surprisingly, that the approximation gets better as the network is trained using SGD, such that the local linearization is almost exact when the network is fully trained. For instance, Figure 4 from Lee et al. [26] shows that a wide residual network and its linearization around initialization behave similarly on Cifar-10. Figure 11 of Mohamadi et al. (https://arxiv.org/abs/2206.12543) further shows that the gaps in test accuracy between various neural network architectures and their local linearization converge to 0 as the neural network is trained under SGD. As we approximate the output of a neural network trained on an additional data points until convergence using a local linearized network, the previous results support that our approximation is also good even for finite width neural networks.

To further demonstrate the quality of the approximation in the active learning (re-training) setting, we provide additional experiment results. Here, we initially train a 1-layer WideResNet on 100, 120, …, 280 random data points from MNIST (the numbers of data points in Figure 1), then consider both SGD training and the NTK approximation with 20 additional random data points. We then compare how often the two network’s predictions agree on a test set, as well as the mean L2 norm of the distance between output predictions.


> initial data points |   100   |   120   |   140    |   160   |   180   |   200   |   220   |   240   |   260   |   280   |
> ----------------|---------|---------|---------|---------|---------|---------|---------|---------|---------|---------|
> prediction agreement        | 0.952 | 0.954  | 0.968 | 0.973 | 0.980 | 0.989 | 0.986 | 0.968 | 0.972  | 0.988  |
> L2 distance           | 0.030 | 0.029 | 0.022 | 0.021 | 0.019 |  0.020 | 0.016 |  0.021 |  0.019  |  0.020  |

This shows that the approximation is quite accurate, disagreeing on at most 5% of test points.

We further compare how, if one uses the two predictions to build MLMOC query batches of size 100, how much those batches would overlap: when training from 100 initial points the batches are 90% overlapping, from 200 initial points they are 92% overlapping, and from 280 initial points they are 93% overlapping. Thus, it seems that our approximation is a good approximation to the computationally infeasible retraining-based method.

---

### Meta-Review · Area_Chair_pQHm · 2022-08-23

**Recommendation:** Accept
**Confidence:** Less certain

**Metareview:**

The paper presents an active learning method which selects samples from an unsupervised pool to be labeled, based on an estimate of the output changes if the selected samples were labeled (thus the name look-ahead). The estimated behavior of the networks after incorporating the selected samples is computed based on the neural tangent kernel approximation to wide neural networks. The authors provided theoretical results for the infinite width case, and also provided practical implementations with local linearizations (with efficient linear algebra operations, which is more straightforward). The method is compared with alternatives to show faster convergence to high quality final model. The authors shall take into account discussions to verify the approximation quality of linearization, and make more rigorous statements to avoid confusions.

**Award:**

No

---

### Decision · Program_Chairs · 2022-09-14

Accept